# BINGO: BOOSTING EFFICIENT REASONING OF LLMS VIA DYNAMIC AND SIGNIFICANCE-BASED REINFORCEMENT LEARNING

## ABSTRACT

Large language models have demonstrated impressive reasoning capabilities, yet they often suffer from inefficiencies due to unnecessarily verbose or redundant outputs. While many works have explored reinforcement learning (RL) to enhance reasoning abilities, most primarily focus on improving accuracy, with limited attention to reasoning efficiency. Some existing approaches introduce direct length-based rewards to encourage brevity, but this often leads to noticeable drops in accuracy. In this paper, we propose BINGO, an RL framework that advances length-based reward design to boost efficient reasoning. BINGO incorporates two key mechanisms: a significance-aware length reward, which gradually guides the model to reduce only insignificant tokens, and a dynamic length reward, which initially encourages elaborate reasoning for hard questions but decays over time to improve overall efficiency. Experiments across multiple reasoning benchmarks show that BINGO improves both accuracy and efficiency. It outperforms the vanilla reward and several other length-based reward baselines in RL, achieving a favorable trade-off between accuracy and efficiency. These results underscore the potential of training LLMs explicitly for efficient reasoning. Our code can be found at https://anonymous.4open.science/r/Bingo-luck-1124.

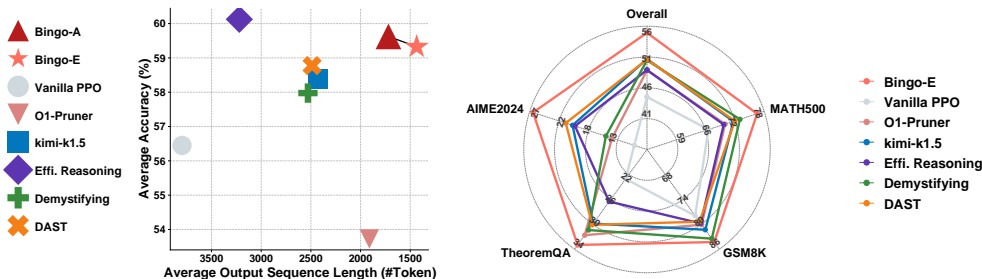

Figure 1: **Performance overview of BINGO and other baselines.** *Left*: Scatter plot of average accuracy versus average response length on four benchmarks (MATH500, GSM8K, TheoremQA, AIME2024) using DeepSeek-R1-Distill-Qwen-1.5B as the base model. Points nearer the top-right corner represent a better balance of accuracy and efficiency. *Right*: Radar chart of length-normalized accuracy for each method. Greater radial distances denote higher efficiency.

## 1 INTRODUCTION

Large language models (LLMs) (OpenAI, 2024; Gunasekar et al., 2023) have demonstrated impressive reasoning capabilities across a variety of tasks, from arithmetic problem solving (Uesato et al., 2022; Hendrycks et al., 2021; Veeraboina, 2023) to commonsense reasoning (Chen et al., 2023). A key observation from recent work is that sufficiently large models can exhibit emergent reasoning abilities, such as chain-of-thought (CoT) reasoning (Wei et al., 2022b), without explicit supervision (Wei et al., 2022a; Suzgun et al., 2022). Despite these successes, a major challenge persists: LLMs often generate unnecessarily verbose or redundant reasoning traces, leading to inefficiencies in computational cost, redundancy, and latency.

Improving reasoning efficiency of LLMs has thus emerged as an important research direction (Qu et al., 2025; Sui et al., 2025; Li et al., 2025; Wang et al., 2025). Prior work in this area can be broadly categorized into supervised fine-tuning (SFT) approaches (Xia et al., 2025; Xu et al., 2025a; Zhang et al., 2025; Kang et al., 2024) and reinforcement learning (RL) approaches (Luo et al., 2025; Team et al., 2025; Arora & Zanette, 2025; Aggarwal & Welleck, 2025; Yeo et al., 2025; Shi et al., 2025). SFT-based methods focus on constructing compressed reasoning traces and training models to imitate them. While these approaches can be effective, they rely on high-quality compressed supervision, which is costly to obtain and often lacks generalizability across diverse tasks. RL-based methods typically introduce length-based rewards that penalize overly long responses to encourage brevity. However, the design of such rewards or penalties in RL-based methods remains underexplored and is often overly simplistic. For example, O1-Pruner (Luo et al., 2025) applies a uniform penalty to all samples, assuming that every response should be shortened. This assumption often leads to performance degradation, as not all reasoning traces are equally verbose—some require more detailed steps to arrive at the correct answer. To address this, other works have proposed more selective penalty strategies, conditioning penalties on sample correctness (Qu et al., 2025; Team et al., 2025; Yeo et al., 2025) or estimated difficulty (Shi et al., 2025). These approaches typically assign stronger penalties to simpler questions and weaker ones to more challenging cases. However, accurately estimating question difficulty remains a fundamental challenge, and unresolved hard questions often lead to unnecessarily long responses, further undermining reasoning efficiency.

Despite growing interest, current designs of length-based rewards remain limited, as they often fail to adequately promote concise reasoning while preserving answer accuracy. For example, prior work has largely overlooked the impact of token-level contributions on the overall efficiency of reasoning. In this work, we approach the problem from a novel perspective grounded in the concept of ***token significance***. Our motivation arises from observed token redundancy in LLMs (Hou et al., 2022; Lin et al., 2025), where many tokens in chain-of-thought (CoT) reasoning contribute little to the final answer. We posit that *not all tokens are equally important for efficient reasoning*—many are insignificant, such as redundant phrases or unnecessary intermediate steps, and can be removed without degrading performance. Existing reward designs often overlook this distinction. In contrast, we introduce a ***significance-aware length reward*** that selectively penalizes only those insignificant tokens which do not meaningfully contribute to the final answer, while preserving essential reasoning steps.

We also observe that effectively handling hard questions is essential for efficient reasoning. Prior work (Muennighoff et al., 2025; Wu et al., 2025) has shown that encouraging extended CoT reasoning can improve performance by enabling deeper exploration, which may help solve more difficult questions. Therefore, it is intuitive to use length as an incentive for hard questions. However, LLMs should not only solve difficult questions accurately but also do so concisely. Applying a static length incentive can lead to unnecessarily long responses, which may still fail to produce correct answers. To address this, we incorporate a ***dynamic length reward*** that adapts over the course of training. This reward is applied to significant tokens in incorrect samples to balance exploration and efficiency. Specifically, it encourages longer reasoning in the early training phase to promote exploration, and gradually shifts toward penalizing excessive length in later stages to promote conciseness.

Building on these insights, we introduce BINGO (Boosting Efficient ReasonING in Policy Optimization), a RL framework that incorporates our two proposed reward mechanisms into standard RL algorithms such as Proximal Policy Optimization (PPO) (Schulman et al., 2017). This enables joint optimization of both reasoning accuracy and efficiency. Extensive experiments across diverse reasoning benchmarks show that BINGO significantly reduces redundant computation while maintaining—or even improving—task accuracy, consistently outperforming strong baselines. As shown in Figure 1, our method delivers substantial gains on both simple and challenging datasets. For example, on the relatively straightforward GSM8K benchmark, it improves accuracy by 1.6 percentage points while reducing response length by 57%. On the more challenging TheoremQA dataset, it achieves a 4.5-point accuracy improvement and a 60% reduction in response length.

In summary, this paper makes the following key contributions:

- **Token Significance Insight.** We introduce the concept of *token significance* in policy optimization, distinguishing between *significant* and *insignificant* tokens in reasoning traces. This insight motivates our *significance-aware length reward*, which explicitly penalizes uninformative tokens while preserving critical reasoning content, enabling more targeted and effective length control.

- **Dynamic Length Control.** We propose a *dynamic length reward* strategy that adjusts the reward signal over the course of training—encouraging longer reasoning in the early stages to foster exploration, and gradually promoting conciseness as the model converges.
- **Efficiency-Oriented RL Framework.** We develop BINGO, a new reinforcement learning framework that integrates both reward strategies. Extensive experiments across multiple reasoning benchmarks, along with comprehensive analyses, demonstrate its effectiveness.

## 2 RELATED WORK

**Reinforcement Learning for Large Language Models.** Reinforcement Learning (RL) (Kaelbling et al., 1996) has emerged as a powerful paradigm for aligning large language models (LLMs) with human preferences. In Reinforcement Learning from Human Feedback (RLHF) (Christiano et al., 2017; Stiennon et al., 2020; Ouyang et al., 2022), the Proximal Policy Optimization (PPO) algorithm (Schulman et al., 2017) is employed alongside human preference data to train a reward model that steers the fine-tuning of LLMs. Building on PPO, subsequent works have proposed improved variants to address its limitations. For instance, GRPO (Shao et al., 2024) improves the stability of reward modeling, while REINFORCE++ (Hu, 2025) focuses on enhancing training efficiency. Beyond alignment, RL has also shown promise in improving the reasoning capabilities of LLMs. Early studies (Lightman et al., 2023; Uesato et al., 2022) demonstrated that reward-guided training can enhance multi-step reasoning performance. More recently, DeepSeek-R1 (DeepSeek-AI et al., 2025) demonstrated that large-scale RL can substantially boost reasoning ability across a wide range of tasks, pointing to a promising direction for future work. RL has also been effectively applied in domain-specific scenarios. For example, DeepRetrieval (Jiang et al., 2025) trains models to reason over search engine interactions for improved information retrieval. Fortune (Cao et al., 2025) applies RL to enhance symbolic table reasoning abilities in LLMs through formula.

**Efficient Reasoning with Large Language Models.** Recent advances have empowered language models to perform strong reasoning via inference-time techniques such as chain-of-thought prompting (Wei et al., 2023; Yao et al., 2023; Cao, 2024; Wang et al., 2023) and post-training (Lightman et al., 2023; Uesato et al., 2022; DeepSeek-AI et al., 2025). More recent work has shifted to optimizing both accuracy and efficiency. Some approaches improve efficiency at inference time, such as token-budget-aware reasoning (Han et al., 2025), or prompting strategies like "reason-without-thinking" (Ma et al., 2025) and chain-of-draft (Xu et al., 2025b). Others apply post-training optimization via supervised fine-tuning (SFT), including TokenSkip (Xia et al., 2025), TwT (Xu et al., 2025a), LightThinker (Zhang et al., 2025), and C3oT (Kang et al., 2024). These SFT methods primarily construct high-quality compressed reasoning paths containing key information, and train the models on them. In parallel, RL-based approaches often improve efficiency by incorporating length controls or penalties into their reward functions. For instance, O1-Pruner (Luo et al., 2025) uses offline length rewards comparing samples against mean lengths. Kimi k1.5 (Team et al., 2025) applies online penalties to correct samples only. Efficient Reasoning (Arora & Zanette, 2025) scales rewards inversely with output length. L1 (Aggarwal & Welleck, 2025) optimizes accuracy under user-defined length constraints. Demystifying (Yeo et al., 2025) uses cosine-based penalties—reducing length for correct outputs while encouraging extended reasoning for incorrect ones. DAST (Shi et al., 2025) employs Token Length Budgets to dynamically adjust reasoning length based on problem difficulty. Building on prior RL-based approaches, we advance length-based reward design to enable LLMs to balance reasoning accuracy with computational efficiency.

## 3 METHODOLOGY

In this section, we introduce the design of the significance-aware length reward and the dynamic length reward, and explain how these two reward mechanisms are integrated into the BINGO framework, as illustrated in Figure 2. All notations are list at Appendix S.

### 3.1 TASK FORMULATION

**Chain of Thought Reasoning.** Let $x$ denote a prompt, and let $y = (y_1, y_2, \ldots, y_n)$ represent the sequence generated by a language model parameterized by $\theta$, where $y_i$ is the $i$-th token in the sequence, and $n$ is the total length of the sequence. Tokens are generated autoregressively from the

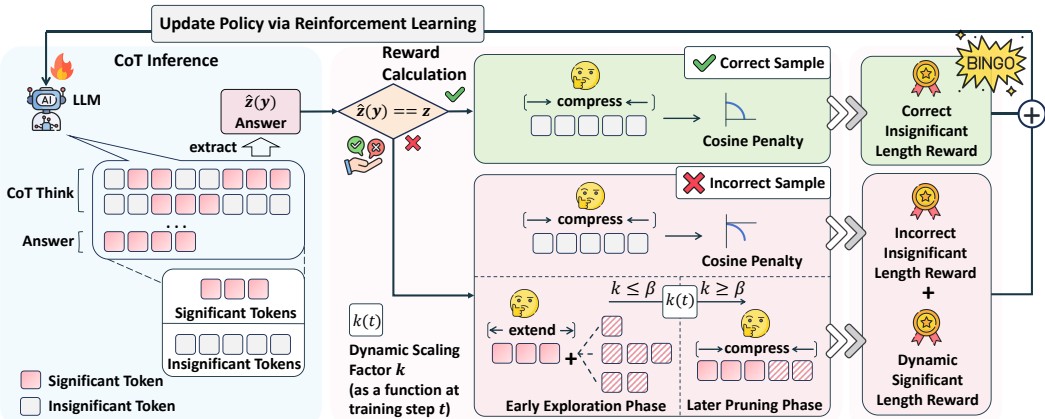

Figure 2: **Illustration of the BINGO framework.** Given a generated CoT trace, the LLM first distinguishes between *significant* and *insignificant* tokens. A dynamic length reward is then computed based on token type and sample correctness. During the early exploration phase of training ($k(t) \geq \beta$), the reward encourages extended reasoning for significant tokens in incorrect samples while penalizing insignificant tokens in all cases. As training progresses ($k(t) < \beta$), the reward shifts toward promoting conciseness by discouraging both significant and insignificant length where appropriate. This two-stage strategy allows the model to first explore broadly and then compress effectively. The aggregated rewards are then used to update the policy via RL, resulting in more accurate and efficient reasoning.

conditional distribution:

$$\pi_\theta(y \mid x) = \prod_{t=1}^{n} \pi_\theta\big(y_i \mid x,\, y_{1:i-1}\big), \tag{1}$$

where the product runs over all tokens in the sequence, with each token $y_i$ (i.e, the action $a_i$) conditioned on the prompt $x$ and the previous tokens $y_{1:i-1}$ (i.e., the state $s_i$). Generation continues until an end-of-sequence (EOS) token is produced, signaling the completion of the response. During this process, the model may produce intermediate reasoning tokens, referred to as a *chain of thought* (CoT) (Wei et al., 2022b), before generating the final answer. Therefore, the full output sequence, denoted as $y$, consists of both the chain of thought and the final answer.

**Optimization Objective of Efficient Reasoning.** The performance of the model in efficient reasoning is assessed along two key dimensions: *accuracy* and *efficiency*.

Accuracy is measured by the *Exact Match* (EM) metric, which evaluates whether the model's final answer matches the ground truth. Let $\hat{z}(y)$ denote the final answer extracted from the model-generated sequence $y$, typically corresponding to its final segment. Let $z$ be the ground-truth answer. Then EM is defined as:

$$\text{EM} = \mathbb{E}_{x \sim \phi} \mathbb{E}_{y \sim \pi_\theta(\cdot \mid x)} \mathbb{1}\left[\hat{z}(y) = z\right], \tag{2}$$

where $\phi$ denotes the distribution over prompts. The indicator function returns 1 if the predicted answer exactly matches the ground truth, and 0 otherwise.

Efficiency is measured by the *response length $L$*, typically defined as the number of tokens $n$ in the generated sequence $y = (y_1, y_2, \ldots, y_n)$. While longer sequences may offer detailed reasoning, they often result in higher computational cost. Thus, reducing unnecessary tokens without harming accuracy is crucial for practical deployment. An ideal model achieves high EM while minimizing the average response length $L$, striking a balance between correctness and conciseness.

### 3.2 SIGNIFICANCE-AWARE LENGTH REWARD

To enhance the efficiency of CoT generation, it is crucial to recognize that not all tokens in a CoT sequence contribute equally to deriving the final answer. *Significant tokens* (such as key concepts, essential terms, or mathematical equations) directly influence the final answer, whereas *insignificant tokens* (including filler words or semantic connectors) contribute little to correctness. Distinguishing

between these two token types is essential for improving CoT efficiency by directing computational resources toward the most informative content.

We leverage LLMLingua-2 (Pan et al., 2024) as an off-the-shelf tool for estimating token-level information content. LLMLingua-2 is a encoder language model, denoted as $\mathcal{M}_e$, specifically trained to assess the significance of individual tokens for text compression. A more detailed discussion of our token significance measurement approach is provided in Appendix E. We define the significance score for each token as follows:

$$S(y_i) = P(y_i \mid \mathbf{y}_{\leq n}; \boldsymbol{\theta}_{\mathcal{M}_e}), \tag{3}$$

where $n$ is the total number of tokens in the output sequence.

Tokens with low importance scores are considered insignificant, while those with high scores are deemed significant. Specifically, we classify tokens as follows:

$$\text{Token } y_i \text{ is } \begin{cases} \text{insignificant}, & \text{if } S(y_i) < \tau, \\ \text{significant}, & \text{if } S(y_i) \geq \tau. \end{cases} \tag{4}$$

We then compute the total number of significant tokens $L^s$ and insignificant tokens $L^{is}$ in the response as:

$$L^s = \sum_{i=1}^{n} \mathbb{1}\left(S(y_i) \geq \tau\right), \quad L^{is} = \sum_{i=1}^{n} \mathbb{1}\left(S(y_i) < \tau\right), \tag{5}$$

where $\mathbb{1}(\cdot)$ is the indicator function, and $\tau$ is a pre-defined threshold. To encourage brevity while maintaining reasoning quality, we introduce a significance-aware length reward that penalizes the excessive use of insignificant tokens through a cosine-based decay:

$$r_{is}(y) = \cos\left(\text{clip}\left(\frac{L^{is}}{L^{is}_{\text{ref}}}, 0, \frac{\pi}{2}\right)\right) + \mathbb{1}[\hat{z}(y) = z] \tag{6}$$

where $L^{is}_{\text{ref}}$ denotes the number of insignificant tokens in a reference response. The cosine function ensures a smooth, non-linear penalty that gradually decreases the reward as $L^{is}$ increases, while the clipping operation bounds the angle to the interval $[0, \frac{\pi}{2}]$, preventing negative rewards. The final reward combines this length-based penalty with an answer reward derived from the EM indicator, ensuring that answer correctness is preserved.

This reward formulation ensures that shorter or equally concise responses—measured in terms of insignificant content—receive higher rewards, while excessively verbose outputs are gently penalized. Notably, our approach preserves natural fluency and coherence in generated text by constraining only the aggregate length of insignificant tokens, without dictating specific token selections or sequences in RL-based training. Compared to standard length-based penalties, our significance-aware approach achieves equal or greater length reductions with less accuracy degradation by selectively penalizing insignificant tokens, as theoretically justified in Appendix F.

### 3.3 DYNAMIC LENGTH REWARD FOR SIGNIFICANT TOKENS

While insignificant tokens are consistently penalized to reduce redundancy, significant tokens warrant a more nuanced approach. In the early stages of training, allowing longer reasoning with significant content can facilitate exploration and support the development of robust problem-solving strategies. However, as training progresses, conciseness becomes increasingly important for improving efficiency.

To accommodate this shift, we introduce a ***dynamic length reward*** for significant tokens that evolves over time based on the model's learning trajectory. This adaptive mechanism is guided by a dynamic scaling factor that captures trends in accuracy and modulates the reward accordingly. Formally, the length-based reward for significant tokens is defined as:

$$r_s(y) = \begin{cases} k \cdot \frac{L^s}{L^s_{\text{ref}}}, & \text{if } k \geq \beta \\ -\alpha \cdot t \cdot \frac{L^s}{L^s_{\text{ref}}}, & \text{if } k < \beta \end{cases} \tag{7}$$

where $L^s$ represents the number of significant tokens in the generated output, $L^s_{\text{ref}}$ is the corresponding value from the reference model, and $k$ is a dynamic scaling factor that reflects the reasoning trend

during training. The training step $t$ begins at 1 and increments gradually when $k$ first falls below the threshold $\beta$, which determines when the model transitions from incentivizing longer significant token lengths to penalizing them. $\alpha$ is a weight that determines the rate of decay in this process. The value of $k$ is estimated by fitting a linear model to recent training steps:

$$k = \frac{\sum_{t=S_a}^{S_b}(t-\bar{t})(acc_t - \overline{acc})}{\sum_{t=S_a}^{S_b}(t-\bar{t})^2} \tag{8}$$

where $acc_t$ denotes the training batch accuracy at training step $t$, $\bar{t}$ and $\overline{acc}$ are the mean step index and mean accuracy over the interval $[S_a, S_b]$. A positive $k$ indicates an upward accuracy trend, suggesting that the model is still in an improvement phase. As training progresses and accuracy plateaus, $k$ approaches zero or becomes negative. The theoretical rationale behind the design of our dynamic length reward schedule is discussed in detail in Appendix G. This dynamic adaptation allows the model to balance early-stage exploration with late-stage compression, fostering reasoning strategies that are both effective and efficient.

## 3.4 Boosting Efficient Reasoning in Policy Optimization

We propose a novel reinforcement learning algorithm, BINGO (**B**oosting Efficient Reason**ING** in Policy **O**ptimization), designed to jointly optimize reasoning performance and efficiency. BINGO extends the reinforcement learning framework—primarily based on Proximal Policy Optimization (PPO) in this work—by introducing two key innovations: a ***significance-aware length reward*** and a ***dynamic length reward***.

As discussed in Section 3.2, we begin by categorizing tokens into *significant* and *insignificant* based on their significance scores. To promote concise yet informative responses, we introduce a cosine-based reward function that adjusts penalties according to the length composition of the response. For correctly answered samples, the reward penalizes only the length of the insignificant portion, reducing verbosity while preserving essential reasoning. For incorrect samples, the reward both penalizes the use of insignificant tokens and encourages the generation of more significant reasoning content.

To balance exploration and efficiency over the course of training, we incorporate a time-dependent mechanism that gradually reduces the incentive for longer responses. As detailed in Section 3.3, this dynamic reward decays as the model converges, shifting the focus from exploration to conciseness.

The overall reward formulation integrates these components into a unified objective:

$$R^{\text{BINGO}}(y) = \begin{cases} \underbrace{\lambda_c \cdot r_{is}(y)}_{\textit{Correct insignificant length reward}}, & \text{if correct,} \\ \underbrace{\lambda_w^{is} \cdot [r_{is}(y) - 1]}_{\textit{Incorrect insignificant length reward}} + \underbrace{\min\left(0, r_s(y) - \lambda_w^s\right)}_{\textit{Dynamic significant length reward}}, & \text{if incorrect.} \end{cases} \tag{9}$$

where the coefficient $\lambda_c$ controls the strength of the penalty applied to correct responses, while $\lambda_w^{is}$ determines the magnitude of the penalty for incorrect ones. The parameter $\lambda_w^s$ serves as a dynamic threshold to balance exploration when the model generates incorrect outputs.

We optimize the policy using the proximal policy optimization objective with the reward $R_{\text{BINGO}}$ defined by Equation 9. The surrogate objective is:

$$\mathcal{J}_{\text{BINGO}}(\theta) = \mathbb{E}_t\left[\min\left(r_t(\theta)\hat{A}_t, \ \text{clip}\left(r_t(\theta), 1-\epsilon, 1+\epsilon\right)\hat{A}_t\right)\right], \tag{10}$$

where:

- $r_t(\theta) = \frac{\pi_\theta(a_t|s_t)}{\pi_{\theta_{\text{old}}}(a_t|s_t)}$ is the importance sampling ratio,

- $\hat{A}_t$ is the advantage estimate at time step $t$, computed via generalized advantage estimation using the final sequence-level reward $R^{\text{BINGO}}$ and the value predictions $V(s_t)$.

- $\epsilon$ is a clipping parameter.

Table 1: **Comparison of different length-based rewards on reasoning benchmarks.** Each method is evaluated using DeepSeek-R1-Distill-Qwen-1.5B as the base model by answer accuracy (Acc, %), response length (Len), and length-normalized accuracy (L-Acc, %). The best performance is highlighted in dark blue, and the second-best in light blue.

| Length-based Reward | MATH500 | | | GSM8K | | | TheoremQA | | | AIME2024 | | |
|---|---|---|---|---|---|---|---|---|---|---|---|---|
| | Acc↑ | Len↓ | L-Acc↑ | Acc↑ | Len↓ | L-Acc↑ | Acc↑ | Len↓ | L-Acc↑ | Acc↑ | Len↓ | L-Acc↑ |
| Vanilla PPO (Schulman et al., 2017) | 81.4 | 2,771 | 66.2 | 85.4 | 1,310 | 78.2 | 32.3 | 4,146 | 22.7 | 26.7 | 6,961 | 10.3 |
| O1-Pruner (Luo et al., 2025) | 74.4 | 991 | 69.8 | 81.4 | 211 | 80.3 | 32.4 | 485 | 31.4 | 26.7 | 5,958 | 13.9 |
| kimi-k1.5 (Team et al., 2025) | 80.4 | 1,692 | 71.6 | 85.4 | 739 | 81.5 | 34.4 | 2,136 | 29.6 | 33.3 | 5,159 | 20.3 |
| Effi. Reasoning (Arora & Zanette, 2025) | 82.6 | 2,395 | 69.5 | 86.4 | 1,155 | 80.0 | 34.8 | 3,560 | 26.2 | 36.7 | 5,771 | 19.9 |
| Demystifying (Yeo et al., 2025) | 80.2 | 1,411 | 73.0 | 86.6 | 548 | 83.6 | 35.1 | 1,976 | 30.6 | 30.0 | 6,183 | 14.9 |
| DAST (Shi et al., 2025) | 81.2 | 1,770 | 71.9 | 82.0 | 456 | 79.6 | 35.2 | 2,325 | 29.8 | 36.7 | 5,400 | 21.4 |
| *Bingo (Ours)* | | | | | | | | | | | | |
| **Bingo-A** | 82.2 | 894 | 77.6 | 87.0 | 563 | 83.9 | 36.8 | 1,648 | 32.9 | 33.3 | 2,943 | 26.7 |
| **Bingo-E** | 80.6 | 779 | 76.7 | 86.7 | 345 | 84.9 | 36.7 | 1,584 | 33.0 | 33.3 | 2,943 | 26.7 |

Therefore, BINGO achieves a favorable trade-off between accuracy and efficiency by maximizing the objective function $\mathcal{J}_{\text{BINGO}}(\theta)$.

## 4 EXPERIMENTS

### 4.1 EXPERIMENTAL SETUP AND EVALUATION METRICS

We fine-tune two reasoning models, *DeepSeek-R1-Distill-Qwen-1.5B* and *DeepSeek-R1-Distill-Qwen-7B*, along with an instruction-tuned model, *Qwen2.5-Math-7B-Instruct*, on the MATH (Hendrycks et al., 2021) training split. Evaluation is conducted on four test sets: MATH500 (Hendrycks et al., 2021), GSM8K (Cobbe et al., 2021), AIME2024 (Veeraboina, 2023), and THEOREMQA (Chen et al., 2023). Among them, MATH500 serves as the in-distribution benchmark, while the others are used as out-of-distribution test sets.

We compare our models with several baselines: the frozen *Base* model (zero-shot), a model fine-tuned with the *Vanilla PPO* algorithm, and five state-of-the-art methods—*DAST* (Shi et al., 2025), *Demystifying* (Yeo et al., 2025), *Efficient Reasoning* (Arora & Zanette, 2025), *Kimi-k1.5* (Team et al., 2025), and *O1-Pruner* (Luo et al., 2025). To isolate and evaluate the effectiveness of different length-based reward designs, we re-implement the core *length-reward components* from all methods within a unified PPO framework. This controlled setup allows direct assessment of whether the reward designs themselves drive performance. For a more detailed description of the experimental settings, please refer to Appendix I. A thorough hyperparameter study can be found in Appendix R.

We report two variants of our model: BINGO-A, the *accuracy-preferred* checkpoint, selected when validation accuracy reaches its peak; and BINGO-E, the *efficiency-preferred* checkpoint, chosen when response length stabilizes during continued training. This dual-reporting strategy enables practitioners to choose a model variant based on their preference for accuracy or efficiency. They may correspond to the same checkpoint.

To evaluate the reasoning efficiency of LLMs in this study, we report not only accuracy (Acc) and response length (Len), but also introduce an additional metric: ***length-normalized accuracy*** (L-Acc). This metric provides a more comprehensive measure of a model's reasoning efficiency by jointly considering correctness and conciseness. It is defined as:

$$\text{L-Acc} = \text{Acc} \times \sqrt{1 - \frac{L}{L_{\max}}}, \tag{11}$$

where $L$ is the average response length and $L_{\max}$ is the maximum allowable length. A detailed definition and theoretical analysis of L-Acc are provided in Appendix H.

### 4.2 PERFORMANCE COMPARISON WITH BASELINE METHODS

**BINGO outperforms existing methods in L-Acc:** As shown in Table 1, both BINGO-A and BINGO-E achieve the highest L-Acc across all four benchmarks, outperforming previous baselines such as Vanilla PPO, Efficient Reasoning, and DAST.

Table 2: **Performance comparison across model scales and types.** Accuracy (Acc), average output length (Length), and length-normalized accuracy (L-Acc, %) on four benchmarks. The best performance is highlighted in dark blue, and the second-best in light blue.

| Method | MATH500 | | | GSM8K | | | TheoremQA | | | AIME2024 | | |
|---|---|---|---|---|---|---|---|---|---|---|---|---|
| | Acc↑ | Len↓ | L-Acc↑ | Acc↑ | Len↓ | L-Acc↑ | Acc↑ | Len↓ | L-Acc↑ | Acc↑ | Len↓ | L-Acc↑ |
| *DeepSeek-R1-Distill-Qwen-1.5B* | | | | | | | | | | | | |
| Base | 63.2 | 3,913 | 45.7 | 73.2 | 2,025 | 63.5 | 18.7 | 5,741 | 10.3 | 16.7 | 7,027 | 6.3 |
| PPO | 81.4 | 2,771 | 66.2 | 85.4 | 1,310 | 78.2 | 32.3 | 4,146 | 22.7 | 26.7 | 6,961 | 10.3 |
| Bingo-A (Ours) | 82.2 | 894 | 77.6 | 87.0 | 563 | 83.9 | 36.8 | 1,648 | 32.9 | 33.3 | 2,943 | 26.7 |
| Bingo-E (Ours) | 80.6 | 779 | 76.7 | 86.7 | 345 | 84.9 | 36.7 | 1,584 | 33.0 | 33.3 | 2,943 | 26.7 |
| *DeepSeek-R1-Distill-Qwen-7B* | | | | | | | | | | | | |
| Base | 82.8 | 3,033 | 65.7 | 85.7 | 1,001 | 80.3 | 37.8 | 4,340 | 25.9 | 40.0 | 6,528 | 18.0 |
| PPO | 88.4 | 1,536 | 79.7 | 92.9 | 918 | 87.5 | 45.4 | 2,709 | 37.1 | 56.7 | 5,857 | 30.3 |
| Bingo-A (Ours) | 88.8 | 1,400 | 80.9 | 92.3 | 371 | 90.2 | 45.2 | 1,908 | 39.6 | 63.3 | 4,670 | 41.5 |
| Bingo-E (Ours) | 87.2 | 1,252 | 80.3 | 91.8 | 366 | 89.7 | 45.0 | 1,693 | 40.1 | 60.0 | 4,011 | 42.9 |
| *Qwen2.5-Math-7B-Instruct* | | | | | | | | | | | | |
| Base | 80.8 | 727 | 70.3 | 95.8 | 331 | 90.3 | 36.8 | 919 | 30.7 | 16.7 | 1,310 | 12.5 |
| PPO | 82.0 | 670 | 72.3 | 96.6 | 305 | 91.5 | 37.6 | 759 | 32.5 | 20.0 | 1,260 | 15.2 |
| Bingo-A (Ours) | 82.6 | 656 | 73.0 | 96.1 | 283 | 91.5 | 37.9 | 598 | 34.0 | 20.0 | 892 | 16.8 |
| Bingo-E (Ours) | 81.6 | 559 | 73.6 | 96.0 | 241 | 92.0 | 37.1 | 552 | 33.5 | 16.7 | 811 | 14.2 |

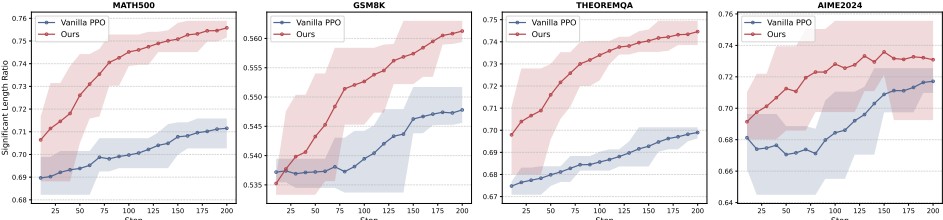

Figure 3: **Significant Length Ratio dynamics during training.** The x-axis indicates training steps, and the y-axis denotes the proportion of significant tokens in the generated responses. Each subplot corresponds to one benchmark evaluated using DeepSeek-R1-Distill-Qwen-1.5B as the base model. The blue curve represents the baseline method (Vanilla PPO), and the red curve represents our approach (Ours).

**BINGO-A improves accuracy while significantly reducing response length:** BINGO-A reduces average response length by up to 68% compared to Vanilla PPO (e.g., 894 vs. 2,771 tokens on MATH500), demonstrating the model's ability to generate concise and correct reasoning steps.

**Existing baselines struggle with the trade-off between accuracy and brevity:** Approaches like Efficient Reasoning produce verbose outputs, while methods such as O1-Pruner overly shorten responses, compromising accuracy.

### 4.3 PERFORMANCE EVALUATION ACROSS VARYING MODEL SCALES

**BINGO achieves the best trade-off between accuracy and response length across different model sizes:** As shown in Table 2, both BINGO-A and BINGO-E consistently outperform all other methods across various model sizes (1.5B and 7B parameters) and benchmarks, achieving the highest L-Acc while maintaining competitive or superior accuracy.

**BINGO-E offers a substantial reduction in response length without sacrificing accuracy:** BINGO-E reduces response length by up to 63% (e.g., 366 vs. 1,001 tokens on GSM8K) compared to the Base model, while also improving accuracy by 6.1 percentage points, demonstrating the model's ability to generate concise and accurate reasoning steps.

### 4.4 ANALYSIS OF SIGNIFICANT VERSUS INSIGNIFICANT TOKEN RATIO

**Significance-aware reward increases the proportion of significant tokens:** As shown in Figure 3, our significance-aware reward consistently improves the significant-token ratio across all datasets. For example, the ratio increases from 0.71 to 0.75 (+4%) on the MATH500. Similarly, the STR improves by about 2% on GSM8K and 5% on TheoremQA, indicating that our method retains the essential reasoning steps while removing redundant or insignificant tokens.

Table 3: **Ablation study on reward components.** Each method is evaluated using DeepSeek-R1-Distill-Qwen-1.5B as the base model by answer accuracy (Acc, %), response length (Len), and length-normalized accuracy (L-Acc, %). Values in parentheses indicate the relative drop in L-Acc compared to BINGO-A. The best performance is highlighted in dark blue, and the second-best in light blue.

| Method | MATH500 | | | GSM8K | | | TheoremQA | | | AIME2024 | | |
|---|---|---|---|---|---|---|---|---|---|---|---|---|
| | Acc↑ | Len↓ | L-Acc↑ | Acc↑ | Len↓ | L-Acc↑ | Acc↑ | Len↓ | L-Acc↑ | Acc↑ | Len↓ | L-Acc↑ |
| **Bingo-A (Ours)** | **82.2** | **894** | **77.6** | **87.0** | 563 | **83.9** | 36.8 | **1,648** | **32.9** | **33.3** | **2,943** | **26.7** |
| Vanilla PPO | 81.4 | 2,771 | 66.2 (-14.7) | 85.4 | 1,310 | 78.2 (-6.8) | 32.3 | 4,146 | 22.7 (-31.0) | 26.7 | 6,961 | 10.3 (-16.4) |
| Significance-Aware Length Reward | 81.4 | 1,734 | 72.3 (-5.3) | 86.7 | 742 | 82.6 (-1.3) | 36.0 | 2,841 | 29.1 (-3.8) | 40.0 | 5,138 | 24.4 (-2.3) |
| w/o Cosine | 78.6 | 1,750 | 69.7 (-7.9) | 85.7 | 509 | 83.0 (-0.9) | 35.3 | 2,414 | 29.7 (-3.2) | 33.3 | 6,454 | 15.4 (-11.3) |
| w/o Significance Separation | 79.8 | 1,666 | 71.2 (-6.4) | 86.6 | 604 | 83.3 (-0.6) | 36.9 | 2,328 | 31.3 (-1.6) | 26.7 | 5,702 | 14.7 (-12.0) |
| w/o Length Incentive | 77.8 | 1,400 | 70.8 (-6.8) | 82.6 | 425 | 80.5 (-3.4) | 35.7 | 1,636 | 32.0 (-0.9) | 30.0 | 4,157 | 21.1 (-5.6) |
| Dynamic Length Reward | 79.0 | 2,204 | 67.5 (-10.1) | 84.3 | 955 | 79.2 (-4.7) | 33.9 | 2,632 | 27.9 (-5.0) | 30.0 | 5,047 | 18.6 (-8.1) |

**Improved reasoning efficiency with richer content:** The increase in significant tokens leads to shorter, more focused chains, as demonstrated by the reductions in response length shown in Table 2. These concise outputs are not only shorter but also contain more meaningful content, resulting in higher raw Acc and L-Acc, reinforcing the effectiveness of our reward strategy.

## 4.5 ABLATION STUDY

**Combining Significance-Aware and Dynamic Length rewards yields the best trade-off:** Table 3 shows that the joint use of both the Significance-Aware and Dynamic Length rewards (BINGO-A) provides the best performance, achieving the highest accuracy and L-Acc across all four benchmarks, while maintaining competitive or superior raw accuracy compared to other methods.

**Removing key reward components degrades performance significantly:** Ablations show that removing any of the reward components leads to noticeable performance drops, particularly in terms of L-Acc. These results confirm the complementary nature of the reward components and their crucial role in optimizing the model's efficiency and accuracy.

## 4.6 ADDITIONAL EXPERIMENTS AND ANALYSIS

**BINGO improves across multiple RL algorithms:** We evaluate the generalizability of our reward design by integrating it into other RL algorithms, including RLOO, GRPO, and Reinforce++. As shown in Appendix K, BINGO variants consistently outperform vanilla ones, achieving superior performance in both accuracy and L-Acc.

**BINGO effectively reduces response length, especially for incorrect samples:** The distribution of response lengths for correct vs. incorrect samples in Appendix O shows that BINGO significantly shortens incorrect sample lengths compared to PPO. Furthermore, Figure 8 in Appendix O illustrates that incorrect samples show a more significant reduction in response length during training, confirming the dynamic reward's effectiveness. Figure 7 in Appendix N further shows that our method consistently reduces response length more than PPO.

**Analysis of significant token ratio and token-level significance visualization:** Appendix L shows that BINGO increases the proportion of significant tokens compared to baselines, while Appendix M provides a token-level significance visualization, demonstrating how our approach retains essential reasoning steps and eliminates redundancy.

**Analysis confirms the effectiveness of dynamic and significance rewards:** The analysis in Appendix P validates that our dynamic and significance rewards balance exploration and efficiency. A case study in Appendix Q further demonstrates the practical impact of BINGO on reasoning efficiency.

## 5 CONCLUSION

In this paper, we introduce BINGO, a RL framework that enhances reasoning efficiency in LLMs. By incorporating significance-aware and dynamic length rewards, BINGO strikes a strong balance between exploration and conciseness, outperforming existing methods across multiple benchmarks.

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

## Contents of Appendix

## A    USE OF LLM ASSISTANCE

In preparing this manuscript, we used large language models (LLMs) solely as writing assistants to help with grammar checking, improving sentence structure and readability, ensuring consistent technical terminology, condensing verbose passages, and formatting citations according to conference guidelines. All research contributions—including ideas, experimental design, data analysis, mathematical derivations, and scientific conclusions—are entirely the authors' original work. The authors take full responsibility for all content presented in this paper.

## B    ETHICS STATEMENT

Our research focuses on improving the efficiency of large language model reasoning through reinforcement learning techniques, which poses no direct ethical concerns regarding human subjects, as no human data collection or experimentation was conducted. All datasets used (MATH, GSM8K, TheoremQA, AIME2024) are publicly available benchmarks with proper citations. We acknowledge the broader implications of more efficient LLM reasoning, including potential dual-use concerns, but emphasize that our contributions aim to reduce computational costs and environmental impact of AI systems. The research was conducted with academic integrity, and all authors have reviewed and agree with the content presented. There are no conflicts of interest to declare.

## C    REPRODUCIBILITY STATEMENT

To ensure reproducibility, we provide comprehensive implementation details throughout the paper. Section 3 describes our complete algorithmic framework, including all hyperparameters and reward formulations. Section 4.1 details our experimental setup, while Appendix I provides comprehensive settings including training and evaluation configurations, dataset settings, optimization parameters, data splits, evaluation metrics, and computational requirements. All experiments use publicly accessible pre-trained models (DeepSeek-R1-Distill-Qwen-1.5B, DeepSeek-R1-Distill-Qwen-7B, and Qwen2.5-Math-7B-Instruct) and datasets available on HuggingFace. Our code is available through an anonymous link in the abstract and as a zip file in the supplemental materials, and will be made publicly available upon acceptance.

## D    LIMITATIONS AND FUTURE WORK

**Limitations.** Despite the promising results of the BINGO framework, there are some limitations that need to be acknowledged:

- **Task-Specific Performance Variability:** While BINGO performs well across several reasoning benchmarks, its performance may vary on more domain-specific or highly complex tasks. Tasks requiring intricate domain knowledge or long-term dependencies may still present challenges.
- **Computational Resources for Training:** The reinforcement learning framework utilized by BINGO requires considerable computational resources for training, which may limit its scalability to larger datasets and more complex tasks.

**Future Work.** Several directions can be explored to improve upon BINGO:

- **Expanding to Diverse Domains and Tasks:** To broaden the applicability of BINGO, it would be beneficial to extend its evaluation to more complex, domain-specific reasoning tasks. This could involve tasks in specialized fields like legal reasoning or advanced scientific modeling.
- **Handling Long-Term Dependencies:** Exploring ways to better handle tasks that require long-term memory or reasoning across large spans of text could make the framework more effective for complex problem-solving scenarios.
- **Improving Training Efficiency:** Future efforts could focus on reducing the computational cost of training by utilizing techniques like transfer learning or distillation, making the framework more accessible for large-scale applications.

# E    DISCUSSION OF TOKEN SIGNIFICANCE MEASUREMENT

As introduced in Section 3.2, we adopt *LLMLingua-2* (Pan et al., 2024) to measure token significance.

A variety of methods have been proposed to mitigate token redundancy in large language models (Hou et al., 2022; Lin et al., 2025), including prompt compression techniques (Li et al., 2023a; Pan et al., 2024; Jiang et al., 2023). One intuitive approach, *Selective Context* (Li et al., 2023b), estimates token importance using a semantic confidence score derived from language modeling:

$$S(y_i) = -\log P(y_i \mid y_{<i}; \theta_{M_L}), \tag{12}$$

where $y_i$ is the $i$-th token, and $\theta_{M_L}$ denotes the parameters of a unidirectional language model. This score reflects the model's uncertainty about each token, using randomness as a proxy for importance. However, this method suffers from two main limitations: (1) position bias, where tokens toward the end of a sequence are systematically assigned lower importance, and (2) the architectural constraint of unidirectional models like GPT, which lack access to future context and thus provide a limited view of token-level informativeness.

In contrast, when designing a significance-aware length reward, we leverage *LLMLingua-2* to compute token significance using:

$$S(y_i) = P(y_i \mid \mathbf{y}_{\leq n}; \boldsymbol{\theta}_{\mathcal{M}_e}), \tag{13}$$

where $\boldsymbol{\theta}_{\mathcal{M}_e}$ denotes the parameters of a bidirectional encoder model. Unlike unidirectional architectures, *LLMLingua-2*—built on BERT-like models—leverages both preceding and succeeding context to evaluate each token, thereby alleviating position bias and enabling a more accurate and holistic assessment of informativeness. Therefore, *LLMLingua-2* serves as a more effective off-the-shelf tool for estimating token-level information content. It is a language encoder model specifically trained to assess the significance of individual tokens for the purpose of text compression. Given a generated sequence $y = (y_1, y_2, \ldots, y_n)$, each token $y_i$ receives an importance score $S(y_i)$, which is used to distinguish between *significant* and *insignificant* reasoning steps.

The advantage of using *LLMLingua-2* over *Selective Context* for measuring token significance has also been validated by recent work such as TokenSkip (Xia et al., 2025).

# F    THEORETICAL ANALYSIS OF SIGNIFICANCE-AWARE LENGTH REWARD

**Preliminaries and Notation.** Given a prompt $x$, the policy $\pi_\theta$ generates a chain-of-thought (CoT) sequence $Y = (y_1, \ldots, y_T)$, from which a deterministic decoder produces a final answer $\hat{Z} \in \mathcal{Z}$. To assess the relative informativeness of each token, we compute a *significance score* using *LLMLingua-2* (Pan et al., 2024):

$$S(y_i) = P(y_i \mid Y; \boldsymbol{\theta}_{\mathcal{M}_e}), \tag{14}$$

where $\boldsymbol{\theta}_{\mathcal{M}_e}$ denotes the parameters of a bidirectional encoder model. Unlike unidirectional predictors, *LLMLingua-2* uses both left and right context to provide a holistic estimate of token informativeness. Based on a fixed threshold $\tau$, we partition the sequence as:

$$\mathcal{Y}_{\text{sig}} = \{y_i : S(y_i) \geq \tau\}, \qquad \mathcal{Y}_{\text{insig}} = \{y_i : S(y_i) < \tau\}, \tag{15}$$

where $\mathcal{Y}_{\text{sig}}$ and $\mathcal{Y}_{\text{insig}}$ denote the sets of *significant* and *insignificant* tokens, respectively.

**Motivation for a Mutual Information Proxy.** In principle, each token's importance could be measured by its mutual information with the final answer, $I(y_i; Z^\star)$. However, computing the exact joint distribution $p(y_i, Z^\star)$ is intractable due to the vast generation space and limited supervision. Instead, we employ a proxy that is (i) efficient to compute for each token and (ii) monotonically correlated with $I(y_i; Z^\star)$.

*LLMLingua-2* (Pan et al., 2024) satisfies these requirements by training under an information bottleneck objective:

$$I(T; Y) - \beta \, I(T; Z^\star), \tag{16}$$

where $T$ is the retained subsequence. Tokens with low conditional probability typically carry little additional information about $Z^\star$, while high-probability tokens preserve essential semantics.

**Assumption 1** (Fidelity of the Mutual Information Proxy). There exist constants $c > \varepsilon > 0$ such that

$$I(y_i; Z^\star) \leq \varepsilon \quad (\forall\, y_i \in \mathcal{Y}_{\text{insig}}), \qquad I(y_j; Z^\star) \geq c - \varepsilon \quad (\forall\, y_j \in \mathcal{Y}_{\text{sig}}). \tag{17}$$

**Lemma 1** (Bounded Accuracy Loss). *Let $\hat{Z}_{full}$ denote the answer decoded from the full CoT, and $\hat{Z}_{sig}$ the answer decoded after removing $\mathcal{Y}_{\text{insig}}$. Under Assumption 1, the increase in error probability is bounded:*

$$\left| \Pr[\hat{Z}_{sig} \neq Z^\star] - \Pr[\hat{Z}_{full} \neq Z^\star] \right| \leq \varepsilon. \tag{18}$$

*Proof.* Let $Y = (y_1, \ldots, y_T)$ and $Y_{\text{sig}} = Y \setminus \mathcal{Y}_{\text{insig}}$. By the chain rule:

$$I(Y; Z^\star) = I(Y_{\text{sig}}; Z^\star) + \sum_{y_i \in \mathcal{Y}_{\text{insig}}} I(y_i; Z^\star \mid Y_{<i}), \tag{19}$$

and each term in the sum is at most $\varepsilon$. Therefore,

$$I(Y; Z^\star) - I(Y_{\text{sig}}; Z^\star) \leq T\varepsilon, \tag{20}$$

and by Fano's inequality, this gap translates into an error increase of at most $\varepsilon$. $\square$

**Definition 1** (General vs. Significance-Aware Length Reward). For a generated trace $Y$, define two reward functions:

$$R_{\text{len}}(Y) = \mathbb{1}[\hat{Z}(Y) = Z^\star] - \lambda\, |Y|, \tag{21}$$

$$R_{\text{sig}}(Y) = \mathbb{1}[\hat{Z}(Y) = Z^\star] - \lambda\, |\mathcal{Y}_{\text{insig}}|. \tag{22}$$

Here, $R_{\text{len}}$ penalizes total length, while $R_{\text{sig}}$ penalizes only insignificant tokens.

**Theorem 1** (Benefit of the Significance-Aware Reward). *Let $\pi_\theta$ be updated by a single PPO step using either reward, with the same coefficient $\lambda > 0$. If*

$$\lambda > \frac{\varepsilon}{\mathbb{E}_{\pi_\theta}[|\mathcal{Y}_{\text{sig}}|]}, \tag{23}$$

*then*

$$\mathbb{E}_{\pi_\theta}[R_{\text{sig}}(Y)] > \mathbb{E}_{\pi_\theta}[R_{\text{len}}(Y)]. \tag{24}$$

*Proof.* Lemma 1 implies

$$\mathbb{E}[R_{\text{sig}}] - \mathbb{E}[R_{\text{len}}] = \lambda\, \mathbb{E}[|\mathcal{Y}_{\text{sig}}|] - \Delta_{\text{acc}}, \tag{25}$$

where

$$0 \leq \Delta_{\text{acc}} \leq \varepsilon. \tag{26}$$

Under the stated bound on $\lambda$, the difference is strictly positive. $\square$

**Practical Implication.** The significance-aware reward achieves the same or greater length reduction with provably smaller accuracy degradation than a general length reward. By selectively penalizing insignificant tokens, it still encourages conciseness while maintaining fidelity. With *LLMLingua-2* providing a fast proxy for token–answer informativeness, this reward design supports both principled and practical optimization for efficient reasoning.

# G    THEORETICAL DISCUSSION OF DYNAMIC LENGTH REWARD

We provide a theoretical discussion of the motivation for our dynamic length reward schedule by addressing three key questions:

1. Why does encouraging longer chains of thought (CoT) during early training help exploration?

2. Why does applying a fixed length penalty throughout training limit performance?

3. Why does dynamically flipping the reward from positive to negative upon convergence yield better accuracy–efficiency trade-offs?

**1. Longer CoT Enables Richer Exploration.**

Let $P_t(L)$ denote the model's distribution over output length $L$ at training step $t$. Define the expected accuracy given length $L$ as

$$\mathrm{Acc}(L) \; = \; \Pr[\hat{Z} = Z^\star \mid L(Y) = L], \tag{27}$$

where $\hat{Z}$ is the predicted answer and $Z^\star$ the ground-truth. Empirically, $\mathrm{Acc}(L)$ follows a saturating "S-curve":

$$\frac{d}{dL}\,\mathrm{Acc}(L) > 0 \quad \text{for } L < L^\star, \qquad \frac{d}{dL}\,\mathrm{Acc}(L) \approx 0 \quad \text{for } L \geq L^\star, \tag{28}$$

where $L^\star$ is the length at which accuracy saturates. The expected accuracy at step $t$ is

$$\mathrm{Acc}_t = \sum_L P_t(L)\,\mathrm{Acc}(L). \tag{29}$$

Shifting probability mass toward longer CoT (up to $L^\star$) thus increases $\mathrm{Acc}_t$, since longer reasoning expands exploration and raises the chance of discovering correct solution patterns.

**Takeaway.**    Rewarding longer CoT early boosts exploration and accelerates convergence toward high accuracy.

**2. Static Length Penalty Causes Premature Compression.**

Consider a fixed penalty $\lambda > 0$, yielding reward

$$J_{\mathrm{static}}(L) = \mathrm{Acc}(L) - \lambda L. \tag{30}$$

The optimal length $L_s$ under this objective satisfies

$$\frac{d}{dL}\,\mathrm{Acc}(L)\big|_{L=L_s} = \lambda. \tag{31}$$

Since $\frac{d}{dL}\,\mathrm{Acc}(L)$ vanishes for $L \geq L^\star$, any $\lambda > 0$ forces $L_s < L^\star$, implying

$$\mathrm{Acc}(L_s) < \mathrm{Acc}(L^\star). \tag{32}$$

Thus the model truncates its CoT before accuracy has fully converged.

**Takeaway.**    Static penalties enforce efficiency too early, sacrificing potential accuracy gains.

**3. Dynamic Penalty Supports a Two-Phase Curriculum.**

We introduce a time-dependent penalty $\lambda_t$:

$$\lambda_t = \begin{cases} \gamma, & t < t_0 \quad \text{(exploration phase)}, \\ \alpha\,(t - t_0), & t \geq t_0 \quad \text{(compression phase)}, \end{cases} \tag{33}$$

where $\gamma < 0$, and $t_0$ is the step at which validation accuracy stabilizes, i.e.,

$$\Delta\mathrm{Acc}_t = \frac{\mathrm{Acc}_t - \mathrm{Acc}_{t-\Delta}}{\Delta} < \beta. \tag{34}$$

**Phase I (Exploration).** During early training, we set $\lambda_t < 0$, effectively turning the penalty into a bonus:

$$J(L) = \mathrm{Acc}(L) - \lambda_t L, \quad \text{with } -\lambda_t > 0,$$

which encourages longer outputs. Since $\mathrm{Acc}(L)$ increases with $L$ up to $L^\star$, this promotes

$$L_t \to L^\star, \qquad \mathrm{Acc}_t \to \mathrm{Acc}(L^\star). \tag{35}$$

**Phase II (Compression).** As training progresses, $\lambda_t$ transitions from negative to positive and increases gradually. When $\lambda_t > 0$, the derivative of the reward at $L^\star$ is

$$\frac{d}{dL}\big[\mathrm{Acc}(L) - \lambda_t L\big]\bigg|_{L=L^\star} = \frac{d}{dL}\,\mathrm{Acc}(L^\star) - \lambda_t < 0, \tag{36}$$

so extending beyond $L^\star$ reduces reward. The policy thus shortens to a new equilibrium $L_d$:

$$\frac{d}{dL}\,\mathrm{Acc}(L)\big|_{L=L_d} = \lambda_t, \quad L_d < L^\star, \quad \mathrm{Acc}(L_d) \approx \mathrm{Acc}(L^\star). \tag{37}$$

**Comparison to Static Penalty.** The final dynamic reward is

$$J_{\mathrm{dyn}} = \mathrm{Acc}(L_d) - \lambda_T L_d, \tag{38}$$

and under the concavity of $\mathrm{Acc}(L)$ one can show

$$J_{\mathrm{dyn}} - J_{\mathrm{static}} = \big[\mathrm{Acc}(L_d) - \mathrm{Acc}(L_s)\big] - \lambda_T(L_d - L_s) \geq 0, \tag{39}$$

i.e., dynamic scheduling yields no worse and often strictly better reward. This holds because for concave functions

$$\mathrm{Acc}(L_d) - \mathrm{Acc}(L_s) \geq \mathrm{Acc}'(L_d)(L_d - L_s), \tag{40}$$

and with $\mathrm{Acc}'(L_d) = \lambda_T$, the inequality follows.

**Efficiency Metric.** Define length-normalized accuracy

$$\text{L-Acc}(L) = \mathrm{Acc}(L)\sqrt{1 - \frac{L}{L_{\max}}}. \tag{41}$$

In practice, dynamic scheduling often achieves similar or higher accuracy with shorter or comparable length, leading to

$$\text{L-Acc}(L_d) > \text{L-Acc}(L_s). \tag{42}$$

**Conclusion.** Our dynamic length reward realizes the curriculum

$$\text{explore freely } (\lambda \leq 0) \;\longrightarrow\; \text{accuracy convergence} \;\longrightarrow\; \text{gradual compression } (\lambda \uparrow). \tag{43}$$

This schedule lets the model reach its accuracy ceiling $\mathrm{Acc}(L^\star)$ before enforcing brevity, achieving better accuracy–efficiency trade-offs than static schemes.

## H  DEFINITION AND THEORETICAL ANALYSIS OF LENGTH-NORMALIZED ACCURACY

**Length-Normalized Accuracy.**

To evaluate reasoning efficiency, we adopt a length-normalized accuracy metric, denoted as L-ACC, which balances correctness with brevity. Formally, it is defined as:

$$\text{L-Acc} = \mathrm{Acc} \times \sqrt{1 - \frac{L}{L_{\max}}}, \tag{44}$$

where $\mathrm{Acc} \in [0, 1]$ denotes exact-match accuracy, $L$ is the number of tokens in the model's response, and $L_{\max}$ is a dataset-specific upper bound on response length. The multiplicative factor penalizes longer outputs in a sub-linear manner, rewarding models that solve problems with fewer tokens.

Specifically, we set $L_{\max} = 8192$ for the two *DeepSeek*-based reasoning models, and $L_{\max} = 3000$ for the *Qwen2.5-Math-7B-Instruct* model, since reasoning-oriented models generally generate longer outputs than instruction-tuned models. The multiplicative factor $\sqrt{1 - \frac{L}{L_{\max}}}$ weights accuracy by a sub-linear penalty on sequence length, so the metric rewards correct solutions that are delivered with fewer tokens. Normalizing by $L_{\max}$ makes the score comparable across datasets of very different scale, while the square-root ensures a smooth, continuous trade-off: the first tokens cut away improve the score more than later ones, mirroring human tolerance for moderate verbosity but aversion to extreme length. When $L = L_{\max}$ the metric collapses to zero, preventing models from exchanging unbounded length for marginal accuracy gains; when $L = 0$ it reduces to the raw accuracy, preserving credit for perfectly concise answers.

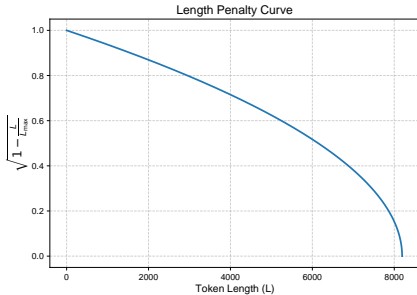

Figure 4: **Penalty curve:** $\sqrt{1 - \frac{L}{L_{\max}}}$.

**Penalty Behavior and Physical Intuition.** The penalty term $\sqrt{1 - \frac{L}{L_{\max}}}$ is continuous, monotonically decreasing, and bounded between 0 and 1. It applies no penalty when $L = 0$, and reduces the reward to zero when $L = L_{\max}$, even if the answer is correct. Crucially, the square-root form introduces diminishing returns: trimming early redundant tokens provides larger gains in L-ACC than removing tokens later in the sequence. This design mirrors human preferences—we tolerate moderate verbosity, but disfavor excessive detail. It also echoes the behavior of L2 regularization, where larger values are penalized more aggressively, while smaller deviations are softly constrained.

**Gradient Analysis.** To understand its optimization implications, we analyze the gradient of the penalty term with respect to $L$:

$$\frac{d}{dL}\left(\sqrt{1 - \frac{L}{L_{\max}}}\right) = -\frac{1}{2L_{\max}} \cdot \left(1 - \frac{L}{L_{\max}}\right)^{-1/2}. \tag{45}$$

This derivative diverges as $L \to L_{\max}$, indicating that long outputs are heavily penalized. In contrast, when $L$ is small, the gradient approaches zero, and the penalty becomes negligible. This behavior encourages models to first eliminate highly redundant tokens, while maintaining stability for shorter outputs.

**Optimization Benefits.** Unlike hard constraints on length, this formulation yields a smooth and differentiable reward signal, making it well-suited for reinforcement learning algorithms such as PPO. It provides stable guidance throughout training and enables the model to trade off between accuracy and length in a controlled and interpretable manner. As shown in Figure 4, the penalty curve strongly discourages excessively long outputs while allowing flexibility in moderately verbose cases, contributing to more efficient and human-aligned reasoning. Notably, the curve becomes steep as the response length approaches $L_{\max}$, meaning that small increases in length lead to sharp drops in reward; conversely, it flattens near $L = 0$, where changes in length have only a minor effect on the reward. This property ensures that the model is heavily penalized for extreme verbosity while remaining tolerant of brief explanatory content.

## I    DETAILED SETTINGS OF EXPERIMENTS

**Prompt.** All experiments use the prompt: `"Let's think step by step and output the final answer within \boxed{}."`

**Models.** Our experiments involve a mix of proprietary and open-source models. The models evaluated in this study include:

- **DeepSeek-R1-Distill-Qwen-1.5B (MIT License)**: A fine-tuned model with 1.5 billion parameters, used to evaluate the proposed method.

- **DeepSeek-R1-Distill-Qwen-7B** (**MIT License**): A fine-tuned model with 7 billion parameters, also used to evaluate the proposed method.
- **Qwen2.5-Math-7B-Instruct** (**Apache-2.0 License**): An instruction-tuned model with 7 billion parameters, used to further assess the efficiency and accuracy in reasoning tasks.

**Datasets.** We evaluate our models on several datasets covering both in-distribution (ID) and out-of-distribution (OOD) tasks. The evaluation framework encompasses:

- **MATH** (Hendrycks et al., 2021): A comprehensive training dataset containing 7,500 mathematical problems across various difficulty levels and topics.
- **MATH500** (Hendrycks et al., 2021): A carefully selected 500-problem subset from the MATH test set, serving as our primary in-distribution evaluation benchmark.
- **GSM8K** (Cobbe et al., 2021): Grade school math word problems requiring multi-step reasoning, used for out-of-distribution evaluation on elementary-level mathematics.
- **TheoremQA** (Chen et al., 2023): A challenging dataset requiring theorem application and symbolic reasoning across STEM domains, used for out-of-distribution evaluation.
- **AIME2024** (Veeraboina, 2023): Problems from the prestigious American Invitational Mathematics Examination, representing the most challenging out-of-distribution evaluation.

These datasets are arranged in increasing order of difficulty: GSM8K < MATH500 < THEOREMQA < AIME2024, offering a comprehensive evaluation of models' reasoning capabilities across varying complexity levels, as summarized in Table 4.

Table 4: Overview of datasets used for training and evaluation.

| Type | Dataset | # Train | # Test | Domain | Task Type | Difficulty | Source |
|------|---------|---------|--------|--------|-----------|------------|--------|
| Training | MATH (Hendrycks et al., 2021) | 7,500 | – | Mathematics | Problem Solving | Mixed | Link |
| ID Test | MATH500 (Hendrycks et al., 2021) | – | 500 | Mathematics | Problem Solving | Medium-Hard | Link |
| OOD Test | GSM8K (Cobbe et al., 2021) | – | 1,319 | Elementary Math | Word Problems | Easy | Link |
| | TheoremQA (Chen et al., 2023) | – | 800 | STEM | Theorem Application | Hard | Link |
| | AIME2024 (Veeraboina, 2023) | – | 30 | Competition Math | Advanced Problem Solving | Very Hard | Link |

**Preprocessing and Tokenization.** Each model uses its corresponding tokenizer to process the input sequences. Tokenization ensures compatibility with the model's input structure, using special tokens to denote the start and end of sequences.

**Training Procedure.** All models are trained for a total of 50 epochs using the Proximal Policy Optimization (PPO) algorithm, optimizing for both accuracy and efficiency. The actor and critic models are initialized with the same parameters, and training is conducted with the following hyperparameters: actor learning rate $= 5 \times 10^{-5}$, critic learning rate $= 1 \times 10^{-6}$, mini-batch size $= 512$, and KL-divergence coefficient $= 0.001$. Evaluation is performed periodically at every training step to monitor progress, and the best model checkpoints are selected for final testing.

**Decoding Configurations.** We conduct both training and evaluation under carefully controlled decoding settings. During training, we adopt sampling generation with temperature $= 1.0$, top_k $= -1$, and top_p $= 1.0$ to encourage exploration, following the default configuration of the VERL framework for comparability with prior work. A single response ($n = 1$) is generated per prompt, with the maximum prompt length capped at 1,024 tokens for efficiency. The maximum response length is set to 8,192 tokens for DeepSeek models and 3,000 tokens for Qwen-Math models.

For evaluation, we emphasize efficiency and stability by adopting greedy decoding, consistent with VERL defaults and prior studies (Yeo et al., 2025; Cui et al., 2025). Specifically, evaluation uses greedy decoding with temperature $= 0$, and one response per input ($n = 1$). The same maximum response lengths as in training are applied (8,192 for DeepSeek, 3,000 for Qwen-Math). We also conducted experiments under extended sampling configurations, with comprehensive results presented in Appendix J.

**Evaluation Metrics.** We evaluate the models using the following metrics:

- **Exact Match (EM)**: Measures the proportion of exact matches between the generated output and the ground-truth answer.
- **Response Length (Len)**: Measures the number of tokens in the output sequence.

- **Length-Normalized Accuracy (L-Acc)**: A metric that balances accuracy and efficiency by considering both correctness and response length.

**Baselines.** We compare the BINGO framework with the following baselines:

- **DAST** (Shi et al., 2025): Uses dynamic length penalties based on problem difficulty.
- **Efficient Reasoning** (Arora & Zanette, 2025): Scales down positive rewards to encourage brevity.
- **Kimi-k1.5** (Team et al., 2025): Applies online length penalties.
- **O1-Pruner** (Luo et al., 2025): Applies offline length penalties based on length comparisons with reference sequences.
- **Demystifying** (Yeo et al., 2025): Applies a symmetric penalty strategy for response lengths, encouraging both shorter and more extensive reasoning depending on correctness.

Rather than directly comparing published baselines—which employ diverse frameworks (e.g., SimPO, GRPO) and differ in their on-policy versus off-policy implementations—we isolate and re-implement only the length-based reward components proposed in each work. All these reward designs are integrated into a unified PPO framework. This approach enables a fair comparison focused specifically on the effectiveness of different reward formulations for improving reasoning efficiency.

**Software and Hardware.** The experiments are conducted with Python 3.11, PyTorch v2.4.0, and CUDA 12.8 for model training and inference. We use 4 NVIDIA A100 80GB PCIe GPUs for training the 7B model and 2 NVIDIA H100 80GB PCIe GPUs for training the 1.5B model. For inference, 2 NVIDIA H100 80GB PCIe GPUs are used to accelerate processing.

## J PERFORMANCE UNDER EXTENDED SAMPLING SETTINGS

We conducted additional experiments using sampling decoding to assess the robustness of our approach under more exploratory conditions. These experiments employed an extended configuration with a 32,768-token output limit, three samples per prompt, temperature of 0.6, and top-p of 1.0. We evaluated the base DeepSeek-R1-Distill-Qwen-1.5B model, vanilla PPO, our proposed Bingo method, and selected competitive baselines to ensure comprehensive comparison.

Table 5: **Performance comparison under sampling decoding settings.** Each method is evaluated using DeepSeek-R1-Distill-Qwen-1.5B as the base model with sampling parameters (32,768 token limit, 3 samples, temperature = 0.6, top-p = 1.0). Metrics include answer accuracy (Acc, %), response length (Len), and length-normalized accuracy (L-Acc, %). The best performance is highlighted in dark blue, and the second-best in light blue.

| Method | MATH500 | | | GSM8K | | | TheoremQA | | | AIME2024 | | |
|---|---|---|---|---|---|---|---|---|---|---|---|---|
| | Acc↑ | Len↓ | L-Acc↑ | Acc↑ | Len↓ | L-Acc↑ | Acc↑ | Len↓ | L-Acc↑ | Acc↑ | Len↓ | L-Acc↑ |
| Base | 81.6 | 5,155 | 74.9 | 83.7 | 1,748 | 81.4 | 31.7 | 7,598 | 27.8 | 17.8 | 15,703 | 12.8 |
| Vanilla PPO | 82.3 | 2,694 | 78.8 | 86.5 | 1,050 | 85.1 | 33.4 | 3,616 | 31.5 | 28.9 | 7,389 | 25.4 |
| O1-Pruner | 80.1 | 1,283 | 78.5 | 85.2 | 352 | 84.7 | 34.7 | 1,095 | 34.1 | 28.9 | 4,636 | 26.8 |
| Demystifying | 81.3 | 1,945 | 78.9 | 86.8 | 483 | 86.2 | 35.2 | 1,863 | 34.2 | 30.0 | 5,891 | 27.2 |
| DAST | 83.5 | 2,053 | 80.8 | 84.1 | 375 | 83.6 | 35.4 | 2,954 | 33.8 | 36.7 | 5,072 | 33.7 |
| *Bingo (Ours)* | | | | | | | | | | | | |
| **Bingo-A** | **85.1** | **1,114** | **83.6** | **88.4** | 483 | **87.7** | **37.9** | 1,592 | **37.0** | **38.9** | **3,110** | **37.0** |
| **Bingo-E** | **84.2** | **983** | **82.9** | **88.1** | **217** | **87.8** | **37.7** | **1,004** | **37.1** | **37.6** | **2,817** | **35.9** |

Table 5 presents the accuracy and average response length across four benchmarks under these sampling conditions. The results demonstrate that Bingo maintains its efficiency advantage even with sampling decoding, achieving strong accuracy while generating substantially shorter outputs than all baseline methods. This finding confirms that our reward design effectively promotes concise reasoning regardless of the decoding strategy employed.

Although these extended settings yielded accuracy improvements, they required approximately five times the computational resources and training time compared to greedy decoding. Given this substantial computational overhead, we selected single-response greedy decoding as our primary evaluation protocol to maintain experimental feasibility while still providing meaningful performance assessments. The sampling results presented here validate that our approach remains effective under more computationally intensive conditions.

Table 6: **Comparison of reinforcement learning algorithms on four reasoning benchmarks.** Each method is evaluated using DeepSeek-R1-Distill-Qwen-1.5B as the base model by answer accuracy (Acc, %), response length (Len), and length-normalized accuracy (L-Acc, %). Bingo-based variants consistently outperform their vanilla counterparts across different RL optimizers (PPO, RLOO, GRPO, Reinforce++). Numbers in parentheses show the L-Acc gain over the corresponding vanilla baseline, with **green** indicating improvement.

| Method | MATH500 | | | GSM8K | | | TheoremQA | | | AIME2024 | | |
|---|---|---|---|---|---|---|---|---|---|---|---|---|
| | Acc↑ | Len↓ | L-Acc↑ | Acc↑ | Len↓ | L-Acc↑ | Acc↑ | Len↓ | L-Acc↑ | Acc↑ | Len↓ | L-Acc↑ |
| Base | 63.2 | 3,913 | 45.7 | 73.2 | 2,025 | 63.5 | 18.7 | 5,741 | 10.3 | 16.7 | 7,027 | 6.3 |
| Vanilla PPO | 81.4 | 2,771 | 66.2 | 85.4 | 1,310 | 78.2 | 32.3 | 4,146 | 22.7 | 26.7 | 6,961 | 10.3 |
| **Bingo-PPO** | 82.2 | 894 | 77.6 **(+11.4)** | 87.0 | 563 | 83.9 **(+5.7)** | 36.8 | 1,648 | 32.9 **(+10.2)** | 33.3 | 2,943 | 26.7 **(+16.4)** |
| Vanilla RLOO | 76.8 | 2,413 | 64.5 | 77.3 | 1,588 | 69.4 | 30.0 | 3,162 | 23.5 | 26.7 | 6,025 | 13.7 |
| **Bingo-RLOO** | 78.0 | 1,985 | 67.9 **(+3.4)** | 80.7 | 450 | 78.5 **(+9.1)** | 32.0 | 2,230 | 27.3 **(+3.8)** | 33.3 | 5,583 | 18.8 **(+5.1)** |
| Vanilla GRPO | 76.4 | 2,533 | 63.5 | 77.8 | 804 | 73.9 | 29.2 | 2,946 | 23.4 | 26.7 | 6,096 | 13.5 |
| **Bingo-GRPO** | 79.4 | 1,753 | 70.4 **(+6.9)** | 80.0 | 449 | 77.8 **(+3.9)** | 31.9 | 2,298 | 27.0 **(+3.6)** | 30.0 | 5,886 | 15.9 **(+2.4)** |
| Vanilla Reinforce++ | 76.2 | 2,842 | 61.6 | 82.0 | 1,291 | 75.2 | 28.0 | 3,977 | 20.1 | 30.0 | 6,168 | 14.9 |
| **Bingo-Reinforce++** | 78.4 | 2,070 | 67.8 **(+6.2)** | 81.0 | 640 | 77.8 **(+2.6)** | 33.1 | 2,566 | 27.4 **(+7.3)** | 30.0 | 5,885 | 15.9 **(+1.0)** |

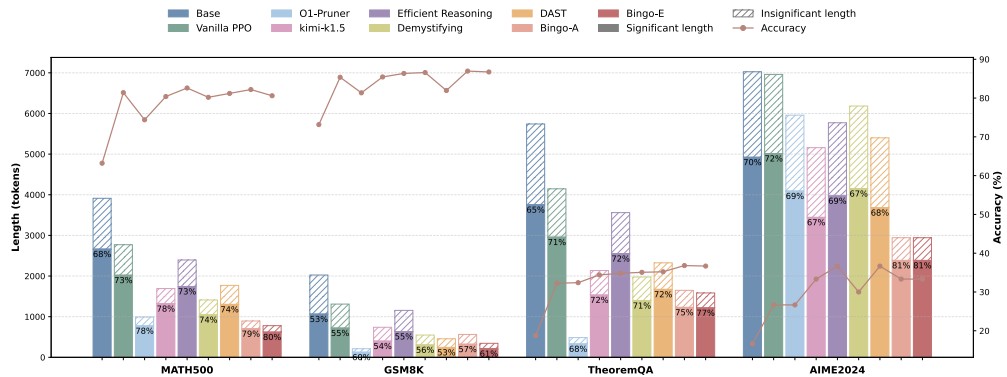

Figure 5: **Length–accuracy results for nine optimization algorithms on four datasets.** Bars show the number of tokens generated using DeepSeek-R1-Distill-Qwen-1.5B as the base model at the checkpoint that yields the reported accuracy (left axis). Each bar is partitioned into significant (dark) and insignificant (light) segments, and the percentage above the bar indicates the share of significant tokens. The solid line (right axis) gives the corresponding answer accuracy. Our methods, Bingo-A and Bingo-E, attain the highest accuracy while using the fewest tokens and achieving the greatest proportion of significant tokens, highlighting their superior reasoning efficiency.

## K PERFORMANCE ACROSS DIFFERENT REINFORCEMENT LEARNING ALGORITHMS

To evaluate the generalizability of our reward design, we integrate it into multiple on-policy reinforcement learning (RL) algorithms, including PPO, RLOO, GRPO, and Reinforce++. As shown in Table 6, Bingo-enhanced variants consistently outperform their vanilla counterparts across all four benchmarks in both accuracy and length-normalized accuracy (L-Acc). Among them, BINGO-PPO delivers the strongest overall performance, achieving the highest or second-highest scores on all datasets while substantially reducing output length. Moreover, the benefits of Bingo extend beyond PPO: each Bingo variant improves L-Acc over its baseline by a clear margin, demonstrating that our reward formulation generalizes well across different policy optimization strategies. These gains are observed not only on in-distribution datasets like MATH500 and GSM8K, but also on more challenging out-of-distribution settings such as AIME2024, highlighting the robustness of our approach. Overall, the results confirm that a principled and learnable length-aware reward offers a consistent advantage across a variety of reasoning tasks and RL algorithms.

**Problem:** Terry eats 2 yogurts a day. They are currently on sale at 4 yogurts for $5.00. How much does he spend on yogurt over 30 days?

**Response of BINGO:**
Okay, so Terry eats 2 yogurt each day, and they're on sale where 4 yogurt cost $5.00. I need to figure out how much he spends over 30 days. Hmm, let me break this down step by step. ... Since 4 yogurt cost $5, and he needs 60 yogurt, which is 15 sets, so 15 times $5 is $75. Terry spends 75.00 dollars on yogurt over 30 days.
(553 tokens with 57% Significant)

**Response of Vanilla PPO:**
Okay, so Terry eats 2 yogurt per day. He's planning to buy them on sale where they're 4 for $5.00. I need to figure out how much he'll spend over 30 days. Wait, let me double-check that. ... Wait, another way to think about it: ... so 12 * $5 = $60. 52 yogurt is 13 sets, so 13 * $5 = $65. 56 yogurt is 14 sets, so 14 * $5 = $70. 60 yogurt is 15 sets, so 15 * $5 = $75. Yep, that's consistent. I think that's solid. So, Terry spends $75.00 on yogurt over 30 days. Terry spends 75 dollars on yogurt over 30 days. (916 tokens with 52% Significant)

**Final Answer:** 75.

Figure 6: **Token-level significance visualization for a sample reasoning task.** Each token is colored based on its predicted significance: red indicates significant tokens (darker = more significant), and blue indicates insignificant tokens (darker = less significant). The response from BINGO (top) is shorter and more concentrated around meaningful reasoning steps, while the Vanilla PPO response (bottom) is longer and contains more exploratory and redundant language. The visualization illustrates how Bingo encourages more efficient and focused reasoning.

## L   ANALYSIS OF SIGNIFICANT TOKEN RATIO

We employ the `DeepSeek-R1-Distill-Qwen-1.5B` model, trained exclusively on the MATH corpus. To evaluate its generalization beyond the training distribution, we test the model on the in-distribution split MATH500 as well as three out-of-distribution (OOD) benchmarks: GSM8K, THEOREMQA, and AIME2024. Figure 5 shows that our approaches, **Bingo-A** and **Bingo-E**, achieve the most favorable length-accuracy trade-off across all four benchmarks.

- **Efficiency at peak accuracy.** At the checkpoints that obtain their highest accuracy, both Bingo variants require only about 20% of the tokens used by the *Base* model on MATH500, with similarly large reductions on GSM8K, THEOREMQA, and AIME2024.
- **Preservation of informative content.** Bingo increases the share of significant tokens to 75–81%, showing that the shortened rationales shed mainly redundant rather than essential reasoning steps.
- **Difficulty-dependent length trends.** Token counts grow with task difficulty: the two harder benchmarks, THEOREMQA and AIME2024, demand considerably longer rationales and yield lower absolute accuracy than MATH500 and GSM8K. Even under these tougher conditions, Bingo still delivers the highest accuracy while generating the fewest tokens.
- **Alleviating the length–accuracy trade-off.** Baselines that compress reasoning without accounting for token importance (e.g., *O1-Pruner*) exhibit marked accuracy declines, whereas Bingo maintains—and in some cases slightly improves—task performance.
- **Robustness across tasks.** The same advantage holds for algebraic, commonsense, formal-logic, and competition-style benchmarks, underscoring the generality of the significance-aware and dynamic length rewards.

These findings confirm that explicitly modeling token significance and adaptively scheduling length rewards enables language models to reason both *accurately* and *efficiently*.

## M   TOKEN-LEVEL SIGNIFICANCE VISUALIZATION

Figure 6 provides a token-level significance visualization for a sample reasoning task. The problem involves calculating the cost of yogurt based on a given sale, and both the BINGO and Vanilla PPO models generate responses to solve the problem. Each token in the generated response is color-coded based on its predicted significance, with red indicating significant tokens and blue representing insignificant ones. Darker shades of red and blue correspond to higher significance levels.

The response from BINGO (top) is notably shorter and more focused on the key reasoning steps, highlighting the model's ability to concentrate on relevant tokens while avoiding unnecessary elabora-

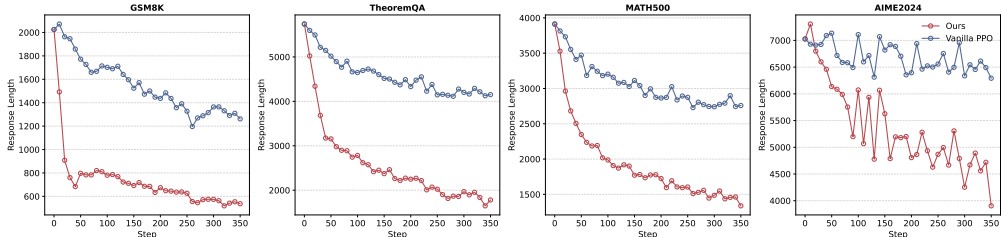

Figure 7: **Response length trends during training across four datasets.** The y-axis shows the number of tokens generated per response using DeepSeek-R1-Distill-Qwen-1.5B as the base model; the x-axis denotes training steps. The red line represents our method, and the blue line corresponds to Vanilla PPO. Across all tasks, our method consistently produces shorter and more stable responses, demonstrating improved reasoning efficiency without compromising task performance.

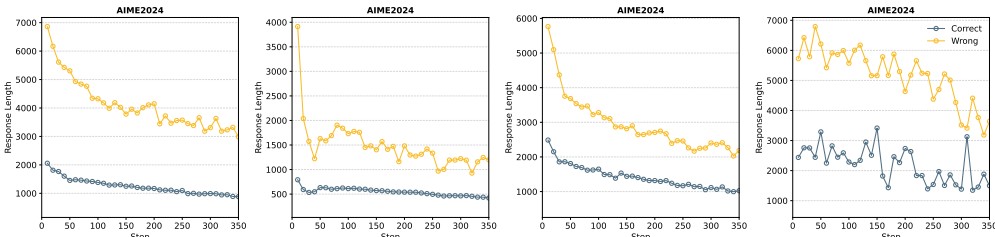

Figure 8: **Response length dynamics for correct vs. wrong samples during training.** The x-axis indicates training steps, and the y-axis denotes response length in tokens for models trained on DeepSeek-R1-Distill-Qwen-1.5B as the base model. The blue line tracks correctly answered samples, while the yellow line tracks incorrectly answered samples. In the early stages, incorrect samples produce substantially longer responses, reflecting the effect of our length-incentive mechanism. As the dynamic length reward gradually diminishes, the response length for incorrect samples falls more sharply than that for correct samples, illustrating the model's adaptive pruning of redundant reasoning steps.

tion. In contrast, the Vanilla PPO response (bottom) is longer, with a higher proportion of redundant and less informative tokens, reflecting a less efficient reasoning process. This visualization clearly demonstrates how BINGO encourages more concise and targeted reasoning, optimizing for both accuracy and efficiency by emphasizing significant steps in the reasoning process.

# N  ANALYSIS OF RESPONSE LENGTHS TRENDS DURING TRAINING

Figure 7 presents the evolution of response length over training steps for Vanilla PPO and our method on four benchmarks. Our approach consistently yields substantially shorter outputs than Vanilla PPO throughout training, demonstrating effective removal of redundant tokens, and converges more smoothly, reflecting robust length regularization. The reduction in response length is most pronounced on the more demanding tasks—MATH500 and AIME2024—where Vanilla PPO produces very long sequences, yet our method maintains a compact reasoning footprint. Importantly, this improvement generalizes across diverse reasoning styles, from arithmetic problems in GSM8K and formal-logic questions in THEOREMQA to academic and competition-style challenges, confirming that our reward design enhances reasoning efficiency without compromising training stability.

# O  ANALYSIS OF RESPONSE LENGTHS DYNAMICS FOR CORRECT VS. WRONG SAMPLES

Figure 8 illustrates how response length evolves for correct and incorrect samples under our approach. In the early phase of training, incorrect samples produce markedly longer outputs than correct ones, demonstrating the impact of our length-incentive mechanism in promoting thorough exploration on

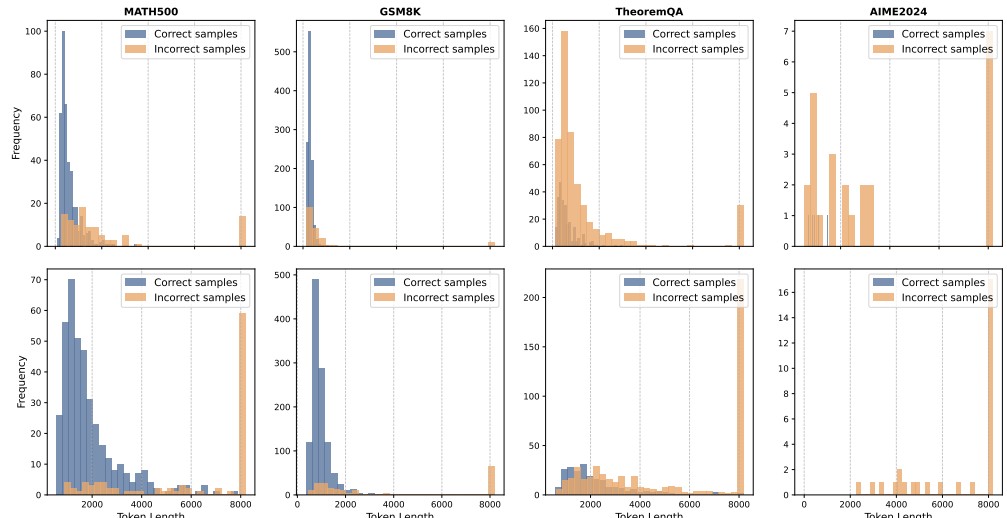

Figure 9: **Distribution of response lengths for correct vs. incorrect samples.** Histograms show the frequency of token lengths in model outputs across four benchmarks using DeepSeek-R1-Distill-Qwen-1.5B as the base model. Each plot compares correct responses (blue) and incorrect responses (orange). The top row corresponds to our method, while the bottom row shows results from Vanilla PPO. Across all datasets, incorrect samples are more likely to produce longer outputs, while correct samples tend to cluster in shorter length ranges. Compared to Vanilla PPO, our method produces a sharper, more compact distribution concentrated in shorter length regions.

challenging cases. As the dynamic length reward takes effect around mid-training, the length for wrong samples declines steeply—outpacing the reduction seen for correct samples—and the gap between the two curves narrows. By later stages, both curves converge toward similarly concise rationales, indicating that the model has learned to apply efficient reasoning uniformly. This behavior confirms that our combination of significance-aware and dynamic rewards not only drives exploration where needed but also enforces brevity once sufficient understanding is achieved, resulting in a balanced, adaptive pruning of redundant tokens.

To examine how response length relates to answer correctness, we compare output length distributions of our method and the Vanilla PPO baseline across four benchmarks using *DeepSeek-R1-Distill-Qwen-1.5B*. As shown in Figure 9, correct responses consistently exhibit shorter lengths than incorrect ones across all tasks. Our method further produces sharply concentrated distributions for correct samples, suggesting more focused and efficient reasoning. In contrast, Vanilla PPO outputs are generally longer and more dispersed, with substantial overlap between correct and incorrect cases. Notably, the length of incorrect samples is substantially reduced compared to Vanilla PPO, suggesting that the dynamic reward mechanism—which gradually penalizes verbosity during training—plays a role in guiding more efficient responses.

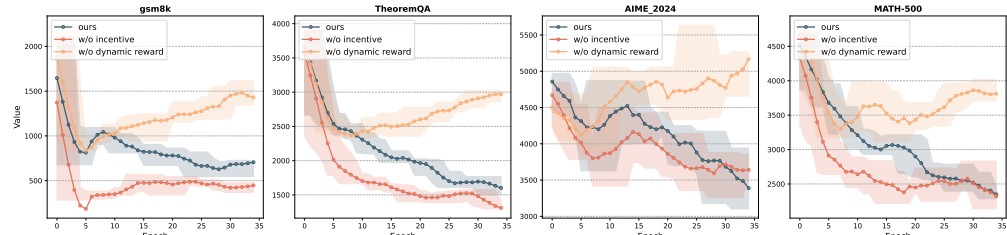

Figure 10: **Effect of Reward Design on Incorrect Response Length.** We visualize the average significant response length of incorrect predictions during training on four benchmarks using DeepSeek-R1-Distill-Qwen-1.5B as the base model. Compared to the variant without incentive, our full method produces longer responses for incorrect samples, suggesting that the significance-aware reward encourages more thorough exploration when the model is uncertain. In contrast, removing the dynamic reward leads to persistently longer outputs, whereas our full method shows a clear reduction in response length over time, confirming the effectiveness of dynamic reward scheduling in promoting concise reasoning. Together, these trends highlight the complementary roles of the two reward components in balancing exploration and efficiency.

# P    ANALYSIS OF INCORRECT RESPONSE LENGTH UNDER DIFFERENT REWARD DESIGNS

To gain deeper insights into the effect of our reward design during training, we analyze the significant response length of incorrect predictions—a proxy for the model's exploratory behavior under uncertainty. As shown in Figure 10, our full method generates longer responses for incorrect samples compared to the version **without incentive** across all four benchmarks, indicating that the significance-aware reward successfully promotes deeper reasoning attempts when the model is uncertain. In contrast, removing the dynamic reward leads to consistently longer outputs throughout training, while our method gradually reduces response length over time. This trend confirms that dynamic reward scheduling effectively encourages concise reasoning as the model matures. Together, these findings highlight the complementary roles of the two reward components: one enhances exploration during failure, and the other improves efficiency during learning progression.

## Q    CASE STUDY

To further assess the reasoning quality and output efficiency of our method, we conduct two case studies comparing the Base model, PPO, and our BINGO framework across two model scales. As shown in Figures 11 and 12, the Base model often produces excessively long and repetitive outputs, sometimes exceeding 5,000 words, and either fails to converge or hides the correct answer deep within verbose reasoning or terminated generations without a final answer. PPO improves conciseness but occasionally generates incorrect outputs or retains unnecessary repetition in intermediate steps. In contrast, BINGO consistently produces correct, well-structured solutions while significantly reducing output length—achieving up to 3x compression over PPO in the 7B setting. These examples demonstrate the effectiveness of our reward design in balancing reasoning depth and brevity, encouraging the model to generate focused and efficient reasoning even on complex tasks.

**Problem: Find the greatest integer less than $(\sqrt{7} + \sqrt{5})^6$. (Do not use a calculator!)**

✗ **Response of Base Model with 2509 words:**
Okay, so I need to find the greatest integer less than $((\sqrt{7} + \sqrt{5})^6/4)$. Hmm, that sounds a bit intimidating at first, but maybe I can break it down step by step. Let me think about how to approach this without a calculator, so I have to rely on algebraic manipulations and maybe some patterns or properties of exponents. First, I remember that expressions like $((\sqrt{a} + \sqrt{b})^n)$ can sometimes be simplified using binomial expansions, but that might get messy for the sixth power. Alternatively, I recall that sometimes pairing such terms can lead to simplifications. Maybe I can consider the conjugate of $(\sqrt{7} + \sqrt{5})$, which is $(\sqrt{7} - \sqrt{5})$, and see if their product is a nice number. Let me compute $\cdots$ Maybe I can find a recurrence relation or some pattern for $(x^n + y^n)$. Let me try computing some lower powers to see if I can spot a pattern. Let's compute $(x + y) : (x + y = (\sqrt{7}+\sqrt{5})+(\sqrt{7}-\sqrt{5}) = 2\sqrt{7})$. Hmm, that's interesting. Let's compute $\cdots$ Okay, so $(x^2+y^2 = 24)$. Let's compute $(x^3 + y^3)$. Maybe I can use the identity $(x^3 + y^3 = (x + y)(x^2 - xy + y^2))$. $\cdots$ But when I computed $(x^6)$ directly, I got approximately 13535.988656. So, which is correct? Wait, perhaps I made a mistake in the direct expansion. Let me check the expansion again. Wait, in the direct expansion, I had: $(x^6 = 6768 + 1144\sqrt{35})$. So, $1144 \times \sqrt{35} \approx 1144 \times 5.916074 \approx$ 6767.988656 So, $\cdots$ So, which is correct? Wait, perhaps I made a mistake in the direct expansion. Let me check the expansion step by step. $\cdots$ So, that's correct. But in the direct expansion, I have $(x^6 \approx 13535.988674)$. So, which is correct? Wait, perhaps I made a mistake in the direct expansion. Let me check the expansion again. Wait, $\cdots$

✗ **Response of PPO with 403 words:**
To find the greatest integer less than $(\sqrt{7} + \sqrt{5})^6$, we can use the property that $(\sqrt{7} + \sqrt{5})^n + (\sqrt{7} - \sqrt{5})^n$ is an integer for any positive integer $n$. Since $\sqrt{7} - \sqrt{5}$ is a small positive number (approximately 0.4142), $(\sqrt{7} - \sqrt{5})^6$ will be a small number close to 0. Let's denote $x = \sqrt{7} + \sqrt{5}$ and $y = \sqrt{7} - \sqrt{5}$. Then, $x + y = 2\sqrt{7}$ and $x - y = 2\sqrt{5}$. However, directly using this might not help. Instead, we know that $x^6 + y^6$ is an integer. Since $y < 1$, $y^6$ is very small, so $x^6$ is slightly less than an integer. Therefore, the greatest integer less than $x^6$ is $x^6 + y^6 - 1$. We need to compute $x^6 + y^6$. To find this, we can use recurrence relations or expand step by step. However, a more efficient way is to compute powers step by step: First, compute $x^2 = (\sqrt{7}+\sqrt{5})^2 = 7+2\sqrt{35}+5 = 12+2\sqrt{35}$. Next, $y^2 = (\sqrt{7}-\sqrt{5})^2 = 7-2\sqrt{35}+5 = 12-2\sqrt{35}$. Then, $x^3 + y^3 = (x + y)(x^2 + y^2 - xy) = 2\sqrt{7} \cdot (24 - 5) = 2\sqrt{7} \cdot 19 = 38\sqrt{7}$. Similarly, $x^3 - y^3 = (x - y)(x^2 + y^2 + xy) = 2\sqrt{5} \cdot (24 + 5) = 2\sqrt{5} \cdot 29 = 58\sqrt{5}$. However, this is getting complex. Instead, we can compute $x^6 + y^6$ using the recurrence relation. Let $S_n = x^n + y^n$. Then, $S_0 = 2$, $S_1 = 2\sqrt{7}$, $S_2 = 24$, $S_3 = 38\sqrt{7}$, $S_4 = 24 \cdot S_2 - S_1^2 = 24 \cdot 24 - (2\sqrt{7})^2 = 576 - 28 = 548$, $S_5 = S_4 \cdot S_1 - S_3 \cdot S_2 = 548 \cdot 2\sqrt{7} - 38\sqrt{7} \cdot 12 = (1096\sqrt{7} - 456\sqrt{7}) = 640\sqrt{7}$, $S_6 = S_5 \cdot S_1 - S_4 \cdot S_2 = 640\sqrt{7} \cdot 2\sqrt{7} - 548 \cdot 24 = 640 \cdot 2 \cdot 7 - 548 \cdot 24 = 8960 - 13152 = -4192$. Thus, $x^6 + y^6 = -4192$. Since $y^6$ is positive, $x^6 = -4192 - y^6$. Therefore, $x^6$ is slightly less than -4192. But this contradicts since $x > 1$. Wait, actually, $y = \sqrt{7} - \sqrt{5} \approx 0.4142$, so $y^6 \approx (0.4142)^6 \approx 0.004$. Thus, $x^6 = -4192 - y^6 \approx -4192.004$, which is negative. This [*Generation Termination without final answer*]

✓ **Response of Bingo with 470 words:**
Okay, so I need to find the greatest integer less than $(\sqrt{7} + \sqrt{5})^6$. Hmm, that sounds like I need to compute $(\sqrt{7} + \sqrt{5})^6$ and then take the floor of that value. But since I can't use a calculator, I have to figure out a way to compute this without approximating. I remember that expressions like $(\sqrt{a} + \sqrt{b})^n$ can sometimes be simplified using binomial expansion or by recognizing patterns. Maybe I can pair $(\sqrt{7} + \sqrt{5})$ with $(\sqrt{7} - \sqrt{5})$ because their product is 7 - 5 = 2, which is nice. Let me denote $x = \sqrt{7} + \sqrt{5}$ and $y = \sqrt{7} - \sqrt{5}$. Then, $x \times y = 2$. I also know that $x + y = 2\sqrt{7}$ and $x - y = 2\sqrt{5}$. Since x and y are conjugates, maybe I can find a recurrence relation for $x^n + y^n$. Let me compute $x^1 + y^1 = 2\sqrt{7} \approx 5.2915$, $x^2 + y^2 = (x + y)^2 - 2xy = (2\sqrt{7})^2 - 2 \times 2 = 28 - 4 = 24$, $x^3 + y^3 = (x + y)(x^2 + y^2 - xy) = 2\sqrt{7} \times (24 - 2) = 2\sqrt{7} \times 22 = 44\sqrt{7} \approx 44 \times 2.6458 \approx 116.315$, $x^4 + y^4 = (x^2 + y^2)^2 - 2(xy)^2 = 24^2 - 2 \times (4) = 576 - 8 = 568$, $x^5 + y^5 = (x + y)(x^4 + y^4) - xy(x^3 + y^3) = 2\sqrt{7} \times 568 - 2 \times (44\sqrt{7}) = 1136\sqrt{7} - 88\sqrt{7} = 1048\sqrt{7} \approx 1048 \times 2.6458 \approx 2767.318$, $x^6 + y^6 = (x^3 + y^3)^2 - 2(xy)^3 = (44\sqrt{7})^2 - 2 \times (8) = 1936 \times 7 - 16 = 13552 - 16 = 13536$, So $x^6 + y^6 = 13536$. But since $y = \sqrt{7} - \sqrt{5}$, which is less than 1 (since $\sqrt{7} \approx 2.6458$, $\sqrt{5} \approx 2.236$, so $y \approx 0.4098$). Therefore, $y^6$ is a very small positive number, less than 1. Therefore, $x^6 = 13536 - y^6$. Since $y^6$ is positive, $x^6 < 13536$ Thus, the integer part of $x^6$ is 13535. Therefore, the greatest integer less than $(\sqrt{7}+\sqrt{5})^6$ is 13535. **Final Answer**: 13535 </think> To find the greatest integer less than $(\sqrt{7} + \sqrt{5})^6$, we denote $x = \sqrt{7} + \sqrt{5}$ and $y = \sqrt{7} - \sqrt{5}$. We know that $x \cdot y = 2$, $x + y = 2\sqrt{7}$, and $x - y = 2\sqrt{5}$. We use the recurrence relation for $x^n + y^n$: - $x^1 + y^1 = 2\sqrt{7}$ - $x^2 + y^2 = 24$ - $x^3 + y^3 = 44\sqrt{7}$ - $x^4 + y^4 = 568$ - $x^5 + y^5 = 1048\sqrt{7}$ - $x^6 + y^6 = 13536$. Since $y = \sqrt{7} - \sqrt{5}$ is less than 1, $y^6$ is a very small positive number. Therefore, $x^6 = 13536 - y^6$, which is slightly less than 13536. Thus, the greatest integer less than $(\sqrt{7} + \sqrt{5})^6$ is 13535.

**Final Answer:** 13535.

Figure 11: Case study under the DeepSeek-R1-Distill-Qwen-1.5B model with three settings: Base, PPO, and Bingo. Blue highlights some redundant and repetitive tokens, while red marks omitted content and the final answer.

1620
1621
1622
1623
1624
1625
1626
1627
1628
1629
1630
1631
1632
1633
1634
1635
1636
1637
1638
1639
1640
1641
1642
1643
1644
1645
1646
1647
1648
1649
1650
1651
1652
1653
1654
1655
1656
1657
1658
1659
1660
1661
1662
1663
1664
1665
1666
1667
1668
1669
1670
1671
1672
1673

**Problem: In how many ways can 8 people sit around a round table if 3 of the people – Pierre, Rosa, and Thomas – all want to sit together? (Two seatings are considered the same if one is a rotation of the other.)**

**✗ Response of Base Model with 5783 words:**

Okay, so I have this function $f(x) = \sqrt{(x-1)}/(x-2)$, and I need to find the smallest possible integer value for x such that $f(x)$ is a real number. Hmm, let me think about how to approach this. First, I remember that for a square root to be real, the expression inside the square root, which is called the radicand, must be greater than or equal to zero. So, in this case, the radicand is (x - 1). That means x - 1 has to be greater than or equal to zero. Let me write that down: $x - 1 \geq 0$ Solving for x, I add 1 to both sides: $x \geq 1$ Okay, so x has to be at least 1. But wait, there's also the denominator in the function, which is (x - 2). For the function to be defined, the denominator can't be zero because division by zero is undefined. So, I need to make sure that x - 2 is not equal to zero. Let me write that condition: ⋯ But x can't be 2, so the next integer after 1 is 2, which is excluded. So, the next integer is 3. wait, is 3 the smallest integer that satisfies both conditions? Let me check. ⋯ wait, if x is 1, then f(x) is 0, which is real. So, 1 is a valid value. But is 1 the smallest integer? ⋯ wait, but hold on, the problem says "the smallest possible integer value for x." So, 1 is the smallest integer in the domain. So, 1 is the answer. But wait, hold on, I just thought of something. The function f(x) is defined as $\sqrt{(x-1)}/(x-2)$. So, for x = 1, it's 0, which is fine. But is 1 the smallest integer? Or is there a smaller integer? But integers go down to negative infinity, but x can't be less than 1 because of the square root. So, 1 is the smallest possible integer. ⋯ So, 1 is the smallest integer in the domain, so that's the answer. But just to make sure, let me think about the function's domain. ⋯ But wait, hold on, I just thought of something else. The problem is asking for the smallest possible integer value for x such that f(x) has a real number value. So, 1 is the smallest integer, but is 1 the smallest integer in the domain? Yes, because x can't be less than 1. But wait, let me think about the function again. If x is 1, f(x) is 0, which is real. So, 1 is acceptable. So, 1 is the answer. ⋯ So, the integers in the domain are 1, 3, 4, 5, etc. So, the smallest integer is 1. Therefore, the [*Generation Termination without final answer*]

**✓ Response of PPO with 437 words:**

To determine the smallest possible integer value for $x$ such that $f(x) = \frac{\sqrt{x-1}}{x-2}$ has a real number value, we need to consider the conditions under which the function is defined. 1. **Numerator Condition**: The expression inside the square root, $x - 1$, must be non-negative because the square root of a negative number is not a real number. $x - 1 \geq 0 \Rightarrow x \geq 1$ 2. **Denominator Condition**: The denominator, $x - 2$, must not be zero because division by zero is undefined. $x - 2 \neq 0 \Rightarrow x \neq 2$ 3. **Combined Domain**: Combining the above conditions, $x$ must be greater than or equal to 1 but not equal to 2. $x \geq 1$ and $x \neq 2$ 4. **Smallest Integer Value**: The smallest integer greater than or equal to 1 is 1. However, we need to check if $x = 1$ satisfies the conditions. - For $x = 1$: $f(1) = \frac{\sqrt{1-1}}{1-2} = \frac{0}{-1} = 0$ Since 0 is a real number, $x = 1$ is valid. Therefore, the smallest possible integer value for $x$ is 1. To determine the smallest possible integer value for $x$ such that $f(x) = \frac{\sqrt{x-1}}{x-2}$ has a real number value, let's analyze the function step by step. 1. **Numerator Condition**: - The expression inside the square root, $x - 1$, must be non-negative because the square root of a negative number is not a real number. $x - 1 \geq 0 \Rightarrow x \geq 1$ 2. **Denominator Condition**: - The denominator, $x - 2$, must not be zero because division by zero is undefined. $x - 2 \neq 0 \Rightarrow x \neq 2$ 3. **Combined Domain**: - Combining the above conditions, $x$ must be greater than or equal to 1 but not equal to 2. $x \geq 1$ and $x \neq 2$ 4. **Smallest Integer Value**: - The smallest integer greater than or equal to 1 is 1. However, we need to check if $x = 1$ satisfies the conditions. - For $x = 1$: $f(1) = \frac{\sqrt{1-1}}{1-2} = \frac{0}{-1} = 0$ Since 0 is a real number, $x = 1$ is valid. Therefore, the smallest possible integer value for $x$ is: 1

**✓ Response of Bingo with 155 words:**

First, I need to determine the domain of the function $f(x) = \frac{\sqrt{x-1}}{x-2}$ to find the smallest integer $x$ for which $f(x)$ is a real number. I know that for the square root to be real, the expression inside must be non-negative. So, $x - 1 \geq 0$, which implies $x \geq 1$. Additionally, the denominator $x - 2$ cannot be zero, so $x \neq 2$. Combining these conditions, the domain is $x \geq 1$ with $x \neq 2$. The smallest integer greater than or equal to 1 is 1, but I need to check if $x = 1$. Plugging in $x = 1$, the denominator becomes $-1$, which is allowed, and the square root is 0. So $f(1) = 0$, which is a real number. Therefore, the smallest possible integer value for $x$ is 1. </think> The function $f(x) = \frac{\sqrt{x-1}}{x-2}$ is defined when the expression inside the square root is non-negative and the denominator is not zero. 1. **Square Root Condition:** $x - 1 \geq 0$. $x \geq 1$. 2. **Denominator Condition:** $x - 2 \neq 0$. $x \neq 2$. Combining these conditions, the domain of $f(x)$ is $x \geq 1$ with $x \neq 2$. The smallest integer value for $x$ that satisfies these conditions is $x = 1$.

**Final Answer:** 1.

Figure 12: Case study under the DeepSeek-R1-Distill-Qwen-7B model with three settings: Base, PPO, and Bingo. Blue highlights some redundant and repetitive tokens, while red marks omitted content and the final answer.

# R  HYPERPARAMETER STUDY

We evaluated several combinations of hyperparameters for BINGO on the GSM8K dataset using DeepSeek-R1-Distill-Qwen-1.5B as the base model. Table 7 reports the accuracy and output length across different settings.

Table 7: Performance of BINGO under different hyperparameter settings on GSM8K.

| $\lambda_c$ | $\lambda_w^{is}$ | $\lambda_w^s$ | $S$ | $\beta$ | $\alpha$ | $\tau$ | Acc. | Len. |
|---|---|---|---|---|---|---|---|---|
| 2 | 2 | 5 | 5 | 2 | 0.5 | 0.5 | 86.6 | 570 |
| 2 | 2 | 5 | 10 | 5 | 0.2 | 0.8 | 86.7 | 585 |
| 5 | 5 | 5 | 10 | 2.5 | 0.4 | 0.6 | 86.9 | 578 |
| **5** | **5** | **5** | **10** | **2.5** | **0.5** | **0.5** | **87.0** | **563** |

**Hyperparameter Definitions:**

- $\lambda_c$: Insignificant Length Reward Weight for Correct Samples.
- $\lambda_w^{is}$: Insignificant Length Reward Weight for Incorrect Samples.
- $\lambda_w^s$: Significant Length Reward Weight for Incorrect Samples.
- $S$: Slope interval for the Dynamic Length Reward.
- $\beta$: Threshold for Training Phase Transition.
- $\alpha$: Decay Factor for Dynamic Length Reward.
- $\tau$: Threshold for Significant Tokens.

As shown in Table 7, the performance of BINGO remains stable, with both accuracy and output length exhibiting only minor fluctuations across the tested hyperparameter ranges. This indicates that the method is robust to hyperparameter choices. Since the last configuration achieves the best overall performance, we fixed these hyperparameters for methods and datasets to ensure consistency and fairness in comparison.

# S    NOTATION TABLE

Table 8 offers a detailed overview of the notations utilized in this paper, along with their respective explanations. It serves as a handy reference to assist readers in grasping the concepts discussed in our work.

Table 8: Notation used throughout the paper

| Notation | Description |
|---|---|
| *General* | |
| $y$ | Sequence of tokens generated by the language model |
| $x$ | Input prompt for the language model |
| $n$ | Total length of the sequence $y$ |
| $y_i$ | $i$-th token in the generated sequence $y$ |
| $\hat{z}(y)$ | Extracted final answer from the generated sequence $y$ |
| $z$ | Ground-truth answer |
| $\mathbb{E}_{\pi_\theta}$ | Expectation over policy $\pi_\theta$ |
| $A(L)$ | Expected accuracy as a function of output length $L$ |
| $L$ | Length of the output sequence generated by the model |
| $L_{\max}$ | Maximum response length in the dataset |
| Acc | Exact match accuracy of the final output |
| L-Acc | Length-normalized accuracy, defined as $\text{Acc} \times \sqrt{1 - \frac{L}{L_{\max}}}$ |
| $S(y_i)$ | Significance score of token $y_i$ |
| $L^s$ | Number of significant tokens in the response |
| $L^{is}$ | Number of insignificant tokens in the response |
| $\tau$ | Threshold for classifying a token as significant or insignificant |
| *Reinforcement Learning* | |
| $\pi_\theta$ | Policy parameterized by $\theta$ |
| $\hat{A}_t$ | Advantage estimate at time step $t$ |
| $r_t(\theta)$ | Importance sampling ratio for policy optimization |
| $R^{\text{BINGO}}$ | Reward function in the BINGO framework |
| $\mathcal{J}_{\text{BINGO}}(\theta)$ | PPO objective with BINGO reward function |
| $\epsilon$ | Clipping parameter in the PPO objective |
| *Rewards and Penalties* | |
| $r_{is}(y)$ | Reward for insignificant tokens in sequence $y$ |
| $r_s(y)$ | Reward for significant tokens in sequence $y$ |
| $\lambda_c$ | Coefficient for penalty on correct responses |
| $\lambda_w$ | Coefficient for penalty on incorrect responses |
| $k$ | Dynamic scaling factor for length reward |
| $\alpha$ | Scaling factor for the decay in dynamic length reward |
| $\beta$ | Threshold for transition between exploration and compression in dynamic reward |
| *Length Penalty* | |
| $L_{\text{ref}}^{is}$ | Reference number of insignificant tokens |
| $L_{\text{ref}}^{s}$ | Reference number of significant tokens |
| $k(t)$ | Dynamic scaling factor for adjusting length reward over time |
| *Miscellaneous* | |
| $\mathcal{M}_e$ | Model used to estimate token significance (LLMLingua-2) |
| $\mathbb{1}[\cdot]$ | Indicator function (1 if true, 0 otherwise) |
| $\mathcal{Y}_{\text{sig}}$ | Set of significant tokens in the sequence $y$ |
| $\mathcal{Y}_{\text{insig}}$ | Set of insignificant tokens in the sequence $y$ |

