# OpenReview forum: "Bingo: Boosting Efficient Reasoning of LLMs via Dynamic and Significance-based Reinforcement Learning"
_ICLR.cc/2026/Conference — ICLR 2026 Conference Withdrawn Submission_

### Official Review · Reviewer_swwJ · 2025-10-31

**Soundness:** 2
**Presentation:** 3
**Contribution:** 2
**Rating:** 2
**Confidence:** 5

**Summary:**

This manuscript presents a novel post-training pipeline designed to enhance the reasoning efficiency of LLMs by explicitly encouraging shorter reasoning paths. The proposed framework introduces two key strategies: (1) a significance-aware length reward, which allocates higher rewards to responses containing fewer insignificant tokens, and (2) a dynamic length reward, which adaptively emphasizes significant tokens at different training stages. Experiments conducted on DeepSeek-R1-Distill-Qwen-1.5B and 7B models demonstrate that these strategies effectively reduce response length while preserving reasoning accuracy.

**Strengths:**

1. A novel algorithm, BINGO, is proposed, incorporating two strategies that assign response rewards based on significant tokens.


2. Experiments verify the effectiveness of BINGO in reducing response length while maintaining reasoning accuracy.

**Weaknesses:**

1. The two proposed strategies based on significant tokens are intuitive but lack theoretical justification or performance guarantees.


2. The proposed BINGO framework relies on additional LLMs (e.g., LLMLingua-2) to identify significant and insignificant tokens. This essentially functions as a form of knowledge distillation, transferring auxiliary knowledge about token significance into the post-trained model. However, incorporating extra LLMs substantially increases computational overhead and reduces training efficiency.


3. The performance improvement achieved by BINGO is marginal. In Table 2, for the 7B models, the accuracy gains over PPO are minimal across most datasets. Moreover, only four methods are compared (Base, PPO, BINGO-A, and BINGO-E), yet the paper highlights two highest scores in blue, which is not appropriate given the limited comparison.


4. In Table 1, within the Len column of TheoremAQ, Q1-Pruner achieves the shortest reasoning length and therefore should be highlighted instead.

**Questions:**

Please see Weaknesses.

---

### Official Review · Reviewer_bVFu · 2025-10-31

**Soundness:** 3
**Presentation:** 3
**Contribution:** 2
**Rating:** 4
**Confidence:** 4

**Summary:**

The paper proposes BINGO, a RL framework that enhances reasoning efficiency in large language models. It introduces two complementary reward functions: a significance-aware length reward, which selectively penalizes insignificant tokens that do not contribute to the final answer. The insignificant tokens are recognized with LLMLingua2. The other is the dynamic length reward, which adaptively shifts from encouraging longer reasoning in early training, and gradually turns to favor concise responses as training progresses. Combined into a PPO framework, the reward assignment depends on answer correctness which penalizes redundant tokens for correct outputs while encouraging deeper reasoning for incorrect ones. Experiments show that BINGO produces some improvement on shorter generation length while maintaining or improving accuracy over prior baselines.

**Strengths:**

1. The paper proposes two rewards in RL training policy towards efficient generation

2. The paper's written flow is clear

3. Experimental results show some advantages in compressing the reasoning model generation while maintaining the accuracy

**Weaknesses:**

I am a bit concerned about using the LLMLingua-2 as a proxy model to evaluate the token significance for reasoning efficiency. I know that LLMLingua2 trains a model to evaluate the token-importance score for prompt compression. However, to my understanding, such "semantic redundancy" is not "logical redundancy" (i.e. too much wait/other thinking pattern in reasoning trace that you are targeting at). In other words, LLMLingua estimates the importance token by token for linguistic redundancy, but those hierarchical reasoning trace is at the step granularity, not the individual token. In the tokenSkip you cite, it clearly demos using the LLMLingua2 will delete some internal tokens, but not globally the less useful thinking step. How much does the LLMLingua2 help to cancel the unnecessary intermediate steps instead of something like repetitive tokens? In the meantime, how did you train the compressor here, did you train a specific compressor for each datasets?


Second is lack of experimental results across models/sizes. The main experimental table comparing Bingo with other baselines only on a 1.5B model in table1. It is not convincing to justify the advantage only on a 1.5B models.

**Questions:**

see weakness

---

### Official Review · Reviewer_PE99 · 2025-11-01

**Soundness:** 1
**Presentation:** 3
**Contribution:** 2
**Rating:** 2
**Confidence:** 4

**Summary:**

Bingo is a RL-based efficient reasoning algorithm with a  significance-aware length reward and dynamic length reward.

**Strengths:**

- Having efficient LRMs is important.
- Distinguish significant and insignificant token looks intuitive.
- Unified implementation of existing methods for a likely more fair comparison.
- Good ablation efforts.

**Weaknesses:**

My major concern is the main experiment setting is too far away from the commonly accepted LRM evaluation setups.

- Improper decoding setting. Most of the experiments are done with temperature 0 and max decoding budget of 8192, this is the opposite of how LRMs are typically used and evaluated (often with a sampling temperature and ~32k budget). The authors did attempt to cover this setting in Appendix J, but it is of only one 1.5B model.

- Limited model coverage, only two LRMs are from deepseek distill, both are below 7B and are Qwens.

- Limited baseline results, baselines are mainly only done on the 1.5B model.

- Missing baselines. Fine-tuning-based baseline like AdaptThink and many of its follow-ups are not compared.

- Limited datasets. Only evaluated on math datasets. Datasets like GSM8K and AIME are not really out-of-distribution of MATH500 as the authors claimed.

**Questions:**

n/a

---

### Official Review · Reviewer_fHhd · 2025-11-01

**Soundness:** 2
**Presentation:** 2
**Contribution:** 2
**Rating:** 4
**Confidence:** 3

**Summary:**

This paper presents BINGO, a reinforcement-learning framework that enables reasoning LLMs to be more concise while maintaining model quality. The authors argue that prior length-based rewards uniformly encourage shorter output length and ignore which tokens actually matter. BINGO introduces token significance and penalizes only insignificant tokens via a cosine-decay length reward combined with exact-match correctness, leaving essential reasoning intact. Significance is estimated using an encoder (LLMLingua-2), which classifies tokens and counts significant vs. insignificant length. To handle hard questions, BINGO adds a dynamic reward on significant tokens: early in training it encourages longer reasoning for exploration, then it gradually penalizes significant length to promote compression. Plugged into PPO, this reframes length control around token-level importance and a learnable schedule. Experiments show consistent gains and large length cuts: e.g., +1.6 EM with 57% shorter outputs on GSM8K and +4.5 EM with 60% shorter outputs on TheoremQA, with improvements generalizing across policy optimizers.

**Strengths:**

1. This paper introduces a novel reward design. The authors propose a significance-aware length reward that selectively penalizes only insignificant tokens (estimated with LLMLingua-2), which is more targeted than blanket length penalties.
2. The experiments demonstrate that the proposed method has clear empirical effectiveness. BINGO achieves the best or near-best accuracy amongst baselines while consuming significantly fewer tokens.

**Weaknesses:**

1. The complex reward design may be susceptible to reward hacking. For example, the trained LLM could learn to generate tokens that LLMLingua-2 deems "significant," rather than genuine reasoning steps that lead to correct solutions.
2. The method depends on a single, task-agnostic, frozen model (LLMLingua-2) to classify token significance. Because this model remains static throughout training, it cannot adapt as the policy evolves. Furthermore, it may underperform in domains it wasn't trained on (e.g., healthcare, finance). An actor-critic style variant, where both the LLM and the significance model are updated adversarially, could mitigate this.
3. The approach introduces additional hyperparameters (the $\lambda$s) compared to existing RL methods , which may be brittle and hard to tune. The paper lacks a sensitivity analysis to assess robustness to these choices.

**Questions:**

1. Can this method be extended to update the significance classifier (LLMLingua-2) during training so it evolves with the policy?
2. How sensitive are the hyper-parameters and their influence on final model quality?

---

### Note · Authors · 2025-12-19

**Comment:**

We thank the reviewers for their valuable feedback and thoughtful comments. We will carefully incorporate the suggestions and revise the paper accordingly for a future submission. Therefore, we respectfully withdraw this submission.

**Withdrawal Confirmation:**

I have read and agree with the venue's withdrawal policy on behalf of myself and my co-authors.